# Systematic analysis of bypass suppression of essential genes

Jolanda van Leeuwen[1,2,*,†] (iD), Carles Pons[3,†], Guihong Tan[2], Zi Yang Wang[2,4], Jing Hou[2], Jochen Weile[2,4,5] (iD), Marinella Gebbia[2,5], Wendy Liang[2], Ermira Shuteriqi[2], Zhijian Li[2], Maykel Lopes[1], Matej Ušaj[2], Andreia Dos Santos Lopes[1], Natascha van Lieshout[2,5], Chad L Myers[6], Frederick P Roth[2,4,5,7] (iD), Patrick Aloy[3,8], Brenda J Andrews[2,4,**] (iD) & Charles Boone[2,4,***] (iD)

## Abstract

Essential genes tend to be highly conserved across eukaryotes, but, in some cases, their critical roles can be bypassed through genetic rewiring. From a systematic analysis of 728 different essential yeast genes, we discovered that 124 (17%) were dispensable essential genes. Through whole-genome sequencing and detailed genetic analysis, we investigated the genetic interactions and genome alterations underlying bypass suppression. Dispensable essential genes often had paralogs, were enriched for genes encoding membrane-associated proteins, and were depleted for members of protein complexes. Functionally related genes frequently drove the bypass suppression interactions. These gene properties were predictive of essential gene dispensability and of specific suppressors among hundreds of genes on aneuploid chromosomes. Our findings identify yeast's core essential gene set and reveal that the properties of dispensable essential genes are conserved from yeast to human cells, correlating with human genes that display cell line-specific essentiality in the Cancer Dependency Map (DepMap) project.

**Keywords** compensatory evolution; gene essentiality; genetic interactions; genetic networks; genetic suppression

**Subject Category** Genetics, Gene Therapy & Genetic Disease

**Mol Syst Biol. (2020) 16: e9828**

## Introduction

Genetic suppression, in its simplest form, occurs when a mutation in one gene overcomes the mutant phenotype associated with mutation of another gene (Botstein, 2015). The general principles underlying this type of genetic interaction are key to our understanding of the genotype-to-phenotype relationship. Frequently, the effect of a mutation is dependent on the genetic background in which it occurs, which complicates the identification of complete sets of causal variants associated with phenotypes, including many common diseases (Nadeau, 2001; Harper *et al*, 2015). In particular, genetic mechanisms driving suppression are relevant to our understanding of genome architecture and evolution. Genetic suppression is also relevant to the resilience of healthy people carrying highly penetrant disease variants and may identify novel strategies for therapeutic intervention (Riazuddin *et al*, 2000; Chen *et al*, 2016b). Mapping genetic interactions, including suppression, in model organisms provides a powerful approach for dissecting gene function and pathway connectivity and for defining conserved properties of genetic interactions that can elucidate genotype-to-phenotype relationships (Costanzo *et al*, 2016; Wang *et al*, 2017; Fang *et al*, 2019).

High-throughput genetic interaction studies derived from synthetic genetic array (SGA) analysis in the budding yeast, *Saccharomyces cerevisiae*, have identified hundreds of thousands of negative and positive genetic interactions, in which the fitness defect of a yeast double mutant is either more or less severe, respectively, than the expected effect of combining the single mutants (Costanzo *et al*, 2010, 2016). These SGA studies involve loss-of-function mutations, either deletion alleles of nonessential genes or temperature-sensitive

1 Center for Integrative Genomics, Bâtiment Génopode, University of Lausanne, Lausanne, Switzerland
2 Donnelly Centre for Cellular and Biomolecular Research, University of Toronto, Toronto, ON, Canada
3 Institute for Research in Biomedicine (IRB Barcelona), The Barcelona Institute for Science and Technology, Barcelona, Spain
4 Department of Molecular Genetics, University of Toronto, Toronto, ON, Canada
5 Lunenfeld-Tanenbaum Research Institute, Sinai Health System, Toronto, ON, Canada
6 Department of Computer Science and Engineering, University of Minnesota-Twin Cities, Minneapolis, MN, USA
7 Department of Computer Science, University of Toronto, Toronto, ON, Canada
8 Institució Catalana de Recerca i Estudis Avançats (ICREA), Barcelona, Spain
*Corresponding author. Tel: +41 21 692 3920; E-mail: jolanda.vanleeuwen@unil.ch
**Corresponding author. Tel: +1 416 978 6113; E-mail: brenda.andrews@utoronto.ca
***Corresponding author. Tel: +1 416 978 6113; E-mail: charlie.boone@utoronto.ca
†These authors contributed equally to this work

(TS) alleles of essential genes with a reduced function. In general, negative genetic interactions are rich in functional information, identifying genes that work together to control essential functions, whereas positive genetic interactions tend to identify more indirect connections (Costanzo *et al*, 2010, 2016). However, the most extreme form of positive genetic interaction is genetic suppression, which often identifies genes within the same general function or pathway (Baryshnikova *et al*, 2010b; Van Leeuwen *et al*, 2016).

Essential genes provide a powerful set of queries for genetic suppression analysis. In *S. cerevisiae*, the set of essential genes was defined by deleting a single copy of each of its ~ 6,000 genes individually in a diploid cell and then testing for viability of haploid deletion mutant offspring (Giaever *et al*, 2002). In total, ~ 18% (~ 1,100) of the ~ 6,000 yeast genes are essential for viability under standard, nutrient-rich growth conditions. Although essential genes tend to play highly conserved roles in a cell (Giaever *et al*, 2002; Costanzo *et al*, 2016), genetic variants can sometimes lead to a rewiring of cellular processes that bypass the fundamental requirement for otherwise essential genes (Dowell *et al*, 2010; Sanchez *et al*, 2019). Spontaneous suppressor mutations can be isolated by selecting for faster growing mutants from large populations of cells that are compromised for the function of an essential gene (Van Leeuwen *et al*, 2016) and can identify bypass suppressors (Liu *et al*, 2015; Chen *et al*, 2016a). Here, we describe the construction of a collection of haploid yeast strains, each carrying a single deletion allele of a different essential gene. We use the collection to test ~ 70% of yeast essential genes for bypass suppression, revealing the set of essential genes that can be rendered dispensable through genetic rewiring, and to discover the general principles of bypass suppression.

# Results

### Global analysis of genetic context-dependent gene essentiality

To systematically identify suppressor mutations that can bypass the requirement of an essential yeast gene, we developed a powerful approach for generating suppressors of essential gene deletion alleles. This method relied on the construction of a collection of haploid "query" strains, each deleted for an essential gene, but viable because of the presence of a TS mutant allele of the same essential gene carried on a plasmid (Appendix Fig S1A, Materials and Methods). To construct these strains, we PCR-amplified TS alleles from available TS strains (Costanzo *et al*, 2016) and cotransformed the PCR product and a linearized plasmid carrying a haploid selection cassette into a diploid yeast strain that was heterozygous for a deletion allele of the corresponding essential gene. The resulting diploid strains carrying an assembled plasmid were sporulated, and haploid progeny carrying the deletion allele of the essential gene and the TS allele on plasmid were selected using the haploid selection cassette present on the plasmid (Appendix Fig S1A, Materials and Methods). The resulting collection contained 1,179 query strains, carrying TS alleles of 728 unique essential genes (~ 70% of all essential yeast genes), with 329 of these genes represented by multiple TS alleles (Dataset EV1).

For each TS query strain, ~ 100–150 million cells were incubated at a range of different temperatures close to the restrictive

temperature of the TS allele, corresponding to 4–6 independent experiments in each case. While these cells often divide slowly to expand the population, the majority will not be able to grow rapidly under these conditions, apart from those that acquire a spontaneous suppressor mutation, which form a distinct colony. The isolation of spontaneous suppressors ensures relatively few genomic mutations, which facilitates the identification of causal single nucleotide polymorphisms (SNPs) through whole-genome sequencing. Cells were subsequently transferred to medium that selected against the plasmid carrying the TS allele of the query gene, to assess for growth in the absence of the essential query gene (Fig 1A). Loss of the plasmid was confirmed using several secondary assays (see Materials and Methods). Ultimately, we isolated a total of 380 suppressor strains that could bypass the requirement for 124 unique essential genes (Dataset EV2).

In the context of previous work, 60 (48%) of our dispensable essential gene set had not been described previously, only 36 (29%) of the genes in our dispensable gene set were previously associated with a bypass suppressor interaction, and for an additional 28 genes, their essentiality is known to be dependent on genetic context but the relevant suppressor gene remains unknown (Dataset EV3). Thirty genes we tested have been described as dispensable essential in the literature, but were not identified as dispensable in our assay (Dataset EV3). For eight of these genes, the published study used a genetic background differing from our S288c model system; 18 genes were identified in a screen in the S288c background but were not characterized for genetic architecture in detail; and only four genes have clearly defined bypass suppressor mechanisms in S288c (Dataset EV3). These four genes may have been missed in our assay due to differences in environmental conditions or slight changes in genetic background between S288c strains from different laboratories. To determine whether testing larger numbers of query mutant cells would have allowed us to identify more rare spontaneous bypass suppressor mutations and potentially expand the list of dispensable essential genes, we compared the number of query mutant cells that were used in the experiments, against the number of identified dispensable essential genes (Fig 1B). This analysis showed that using more query mutant cells in our assay would have been unlikely to identify a substantial number of additional dispensable essential genes under these experimental conditions (Fig 1B). We note that additional essential genes could be dispensable in the presence of specific rare variants that cannot be easily achieved by spontaneous mutation. We conclude that at a minimum, ~ 17% of essential yeast genes are dispensable through spontaneous genomic rewiring.

### Properties of dispensable essential genes

Cellular processes such as translation or protein degradation were rarely found within the dispensable essential gene set, whereas the essentiality of genes involved in nuclear–cytoplasmic transport, signaling, cell cycle progression, cell polarity and morphogenesis, and secretory pathway sorting could frequently be bypassed (Fig 1C). Previous analyses have revealed several properties of essential genes that distinguish them from nonessential genes, including a relative depletion of genes with paralogs, an enrichment of genes encoding protein complex members, and a tendency to be more strongly expressed and to have a higher coexpression degree

(i.e., share similar expression patterns with more other genes) (Jeong *et al*, 2001; Giaever *et al*, 2002; Carlson *et al*, 2006; Michaut *et al*, 2011; Woods *et al*, 2013; Qin *et al*, 2018). However, in agreement with a previous survey (Liu *et al*, 2015), we found that dispensable essential genes tended to behave more like nonessential genes, because compared to other essential genes they (i) were

enriched for genes with paralogs; (i) had a lower coexpression degree; and (iii) were depleted for genes encoding components of protein complexes (Fig 1D). Dispensable essential genes were also enriched for genes encoding membrane-associated proteins. Thus, dispensable essential genes possess distinct gene- and protein-level properties, relative to other essential genes.

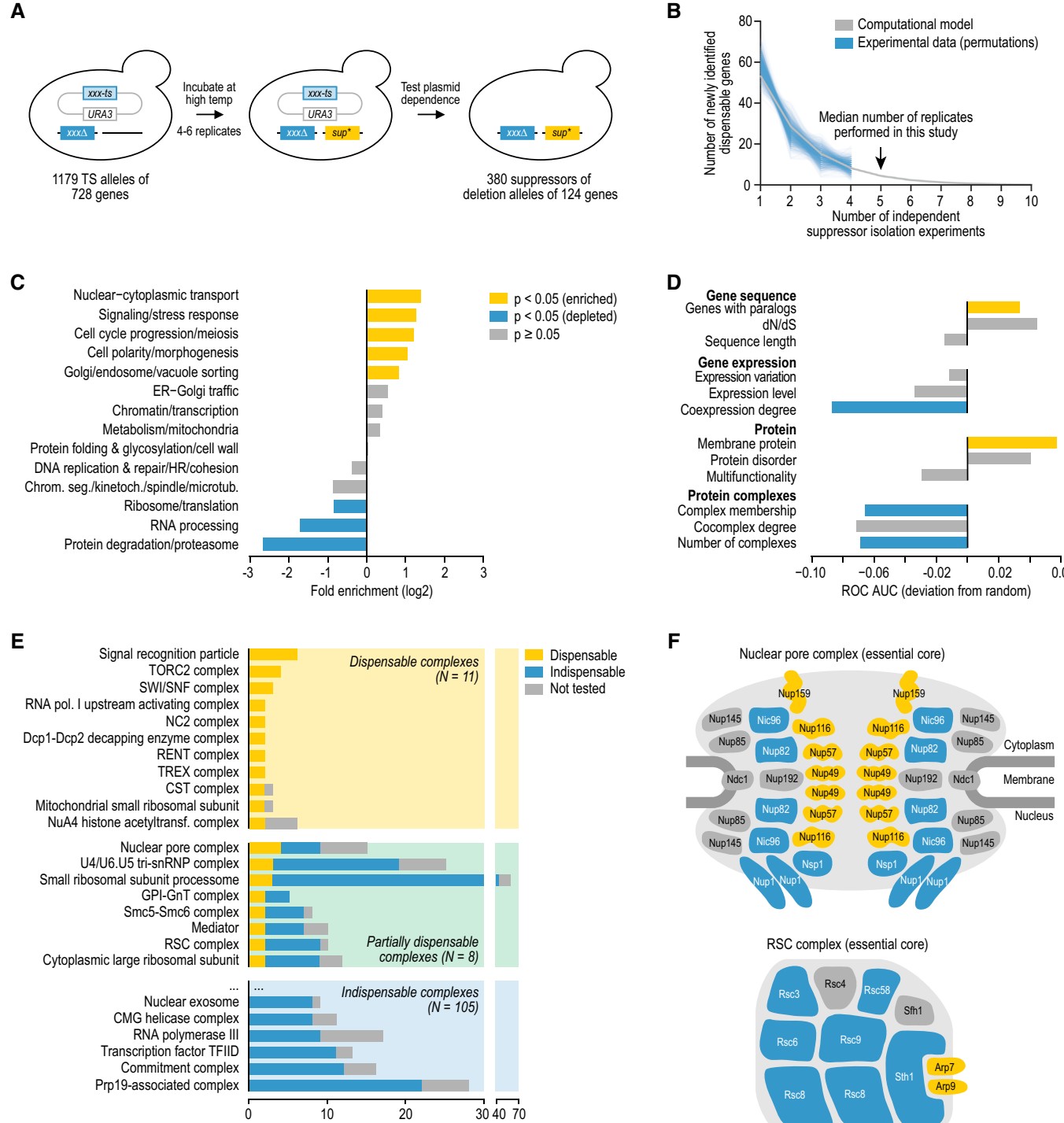

**Figure 1.**

**Figure 1. Properties of dispensable essential genes.**

A  Strategy for isolating bypass suppressors of essential gene deletion mutant alleles.

B  The number of newly identified dispensable essential query genes is plotted against the number of independent suppressor isolation experiments (1 experiment = ~ 25 million query mutant cells). Shown are the 1,000 random permutations of the experimental results and a computational model that was fit to these data. Note that we used at most four independent suppressor isolation experiments per query gene for the random permutations, while a median of five experiments was performed.

C, D  Enrichment of dispensable essential genes among tested essential genes (C) for annotation to a biological functional class and (D) for various gene- and protein-level properties. Fisher's exact or Mann–Whitney *U*-tests were performed to determine statistical significance of the results.

E  Dispensability of essential protein complexes. For each protein complex, the number of subunits encoded by an essential gene is shown, subdivided by their dispensability.

F  The essential subunits of the nuclear pore complex and the chromatin remodeling complex RSC (Hodges *et al*, 2016) are shown. Subunits are color-coded according to their dispensability, using the same color scheme as in (E). Although the essential gene *SEC13* encodes a subunit of the nuclear pore complex, it is not included in the figure as its essentiality results from Sec13's role in another complex, the coat protein complex II (Copic *et al*, 2012).

## Dispensability of essential protein complexes

Most essential genes belong to well-defined protein complexes (Michaut *et al*, 2011), allowing us to investigate gene dispensability within the context of these functional modules. Of 149 protein complexes that contained at least two essential subunits that were tested for their dispensability, 105 complexes (~ 70%) were composed exclusively of indispensable essential genes, such that deletion of none of the essential subunits could be tolerated (Dataset EV4, Fig 1E). These indispensable complexes were part of fundamental cellular machinery, including the proteasome, the exosome, and translation initiation factors, which is consistent with the depletion of dispensable genes among genes involved in protein degradation, RNA processing, and translation (Dataset EV4, and Fig 1C and E). In contrast, for 11 complexes, all tested essential subunits could be bypassed. These dispensable complexes tended to have relatively few essential subunits, and they spanned a variety of biological functions, including protein and mRNA transport (signal recognition particle and TREX complex), signaling (TOR complex 2), and transcriptional regulation (RNA polymerase I upstream activating complex, negative cofactor 2 complex, and RENT complex).

Notably, for eight complexes only a subset of specific essential subunits was dispensable (Fig 1E). These partially dispensable complexes contained a relatively large number of essential subunits and included the nuclear pore complex (15 essential subunits) and the U4/U6.U5 triple small nuclear ribonucleoprotein (25 essential subunits). In several cases, the dispensable essential genes displayed different properties than the indispensable essential genes within the same complex. For example, of genes encoding cytoplasmic large ribosomal subunit proteins, only those with paralogs could be bypassed. In other cases, only members of a specific submodule were dispensable. For instance, subunits of the nuclear pore complex lining the inside of the pore, which are mainly involved in transport specificity, could be bypassed (Onischenko & Weis, 2011; Liu *et al*, 2015), while the structural components tended to be indispensable (Fig 1F). Similarly, for the RSC chromatin remodeling complex, only loss of the actin-related proteins that have a role in the regulation of RSC activity could be bypassed (Szerlong *et al*, 2008), whereas genes encoding subunits with a structural role were indispensable (Fig 1F). Finally, 25 complexes were not further classified (Dataset EV4), because either only one subunit could be bypassed, or the dispensable subunits overlapped with other complexes, so that the observed dispensability may not be related directly to the function of a particular complex.

## Bypass suppressor identification and confirmation

To identify the specific bypass suppressor genes, we performed whole-genome sequencing on the 380 different yeast strains. We identified a median of three rare variants per strain, of which two were often unique nonsynonymous mutations (Datasets EV5 and EV6). Of the 380 suppressor strains we sequenced, 188 (49%) showed changes in genome content, such as aneuploidies (Dataset EV7). To complement the whole-genome sequencing data, we performed high-resolution SGA-based mapping experiments, which can identify the genomic region carrying the suppressor gene (Jorgensen *et al*, 2002), focusing on 89 suppressor strains that had a relatively mild fitness defect and did not carry aneuploidies (Datasets EV2 and EV7, Materials and Methods). For 47 strains, the SGA analysis identified a suppressor locus consisting of ~ 20 unique genes (Datasets EV2 and EV8); however, in another 39 cases, the strains suffered from low spore viability and/or limited sporulation, whereas three strains did not show a clearly identifiable suppressor locus.

Candidate suppressor genes were predicted based on: (i) the presence of a unique nonsynonymous mutation within the candidate gene; (ii) the location of the gene within the genetically mapped suppressor locus; (iii) the reoccurrence of mutations within the same candidate gene in multiple independent suppressor isolates of the same query mutant; and/or (iv) a functional connection between the candidate and query genes. Our approach for identifying candidate suppressor genes on aneuploid chromosomes is described below. All 283 identified candidate suppressor genes were further validated using genetic crosses and complementation assays (Appendix Fig S1B, Dataset EV2, Materials and Methods). The majority (79%) of the tested suppressors were confirmed by at least one of these assays. In total, we identified 141 unique bypass suppression interactions in 259 suppressor strains (Figs 2 and 3A, Dataset EV2). Notably, only 22% of the identified suppression interactions have been reported previously, including both bypass suppression interactions (15%; Dataset EV3) and suppressors of hypomorphic (partial loss-of-function) alleles (7%) (Van Leeuwen *et al*, 2016).

Candidate suppressor gene validation experiments included tetrad analysis of meiotic progeny derived from crossing each suppressor strain to a strain carrying a deletion or hypomorphic allele of the suppressor gene (Appendix Fig S1B). On the basis of this assay and the type of suppressor mutation, one-third of the suppressor mutations appeared to be associated with a gain-of-

function phenotype, while about half appeared to be loss-of-function mutations (Fig 3B; Dataset EV2). The remaining suppressor mutations could not be further classified (Fig 3B).

## Properties of bypass suppressors of essential gene deletion mutants

The essential gene bypass suppressor mutations showed several properties that were similar to the properties of suppressors we previously mapped for nonessential gene deletion mutants that displayed a growth defect (Van Leeuwen *et al*, 2016). For example, the bypass suppressors and their corresponding essential query genes were often annotated to the same biological process (Fig 2) and were enriched for gene pairs that were coexpressed, shared GO annotations, or encoded colocalized proteins or members of the same pathway or complex (Fig 3C). Missense suppressor mutations of essential gene deletion alleles were frequently predicted to be deleterious, often occurred at protein–protein interaction interfaces, and were depleted in disordered protein regions (Appendix Fig S2A). These general findings are consistent with and extend previous findings made with other types of query genes and alleles (Van

Leeuwen *et al*, 2016). However, in contrast to the suppressor mutations of nonessential deletion mutants (Van Leeuwen *et al*, 2016), suppressors of essential gene deletion mutants often occurred in other essential genes (Fig 3D). Suppressor mutations in essential genes frequently involved gain-of-function or overexpression events (73 % of essential suppressor genes). For instance, the lethality associated with a deletion allele of *NTF2*, which encodes a nuclear envelope protein, can be suppressed by increasing the copy number of *GSP1*, which encodes an essential Ran GTPase that controls nucleocytoplasmic transport in collaboration with Ntf2 (Fig 3E). An example of a gain-of-function missense suppressor mutation involves the bypass of actin-related proteins Arp7 and Arp9, which have a role in the regulation of RSC chromatin remodeling complex activity, by specific point mutations in the gene encoding the catalytic RSC ATPase subunit, Sth1 (Dataset EV2, Fig 1F). All eight identified missense mutations cluster within 15 amino acids inside the post-helicase-SANT-associated domain of Sth1 and may increase Sth1 ATPase activity in the absence of ARP proteins (Szerlong *et al*, 2008).

We previously established a classification system to assign suppression interactions to distinct mechanistic categories (Van

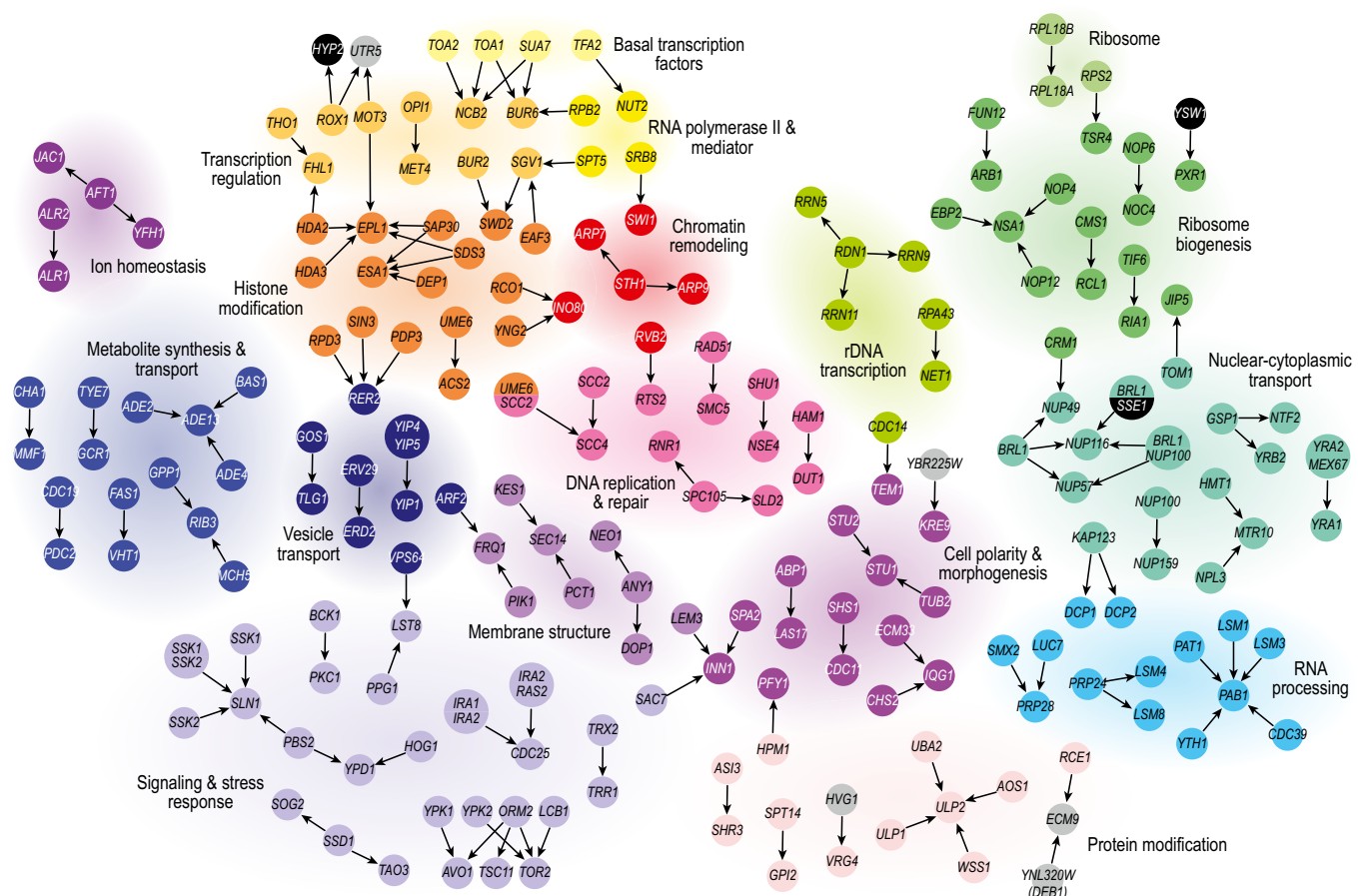

**Figure 2.  Network of bypass suppression interactions.**

Bypass suppressor interactions are represented as arrows that point from the bypass suppressor gene to the essential query gene. Nodes are colored and grouped based on the function of the gene(s). Gray nodes indicate genes that are poorly characterized, whereas black nodes highlight genes with functions that are not otherwise represented in the figure. Complex suppression interactions involving two suppressor genes are represented by larger nodes.

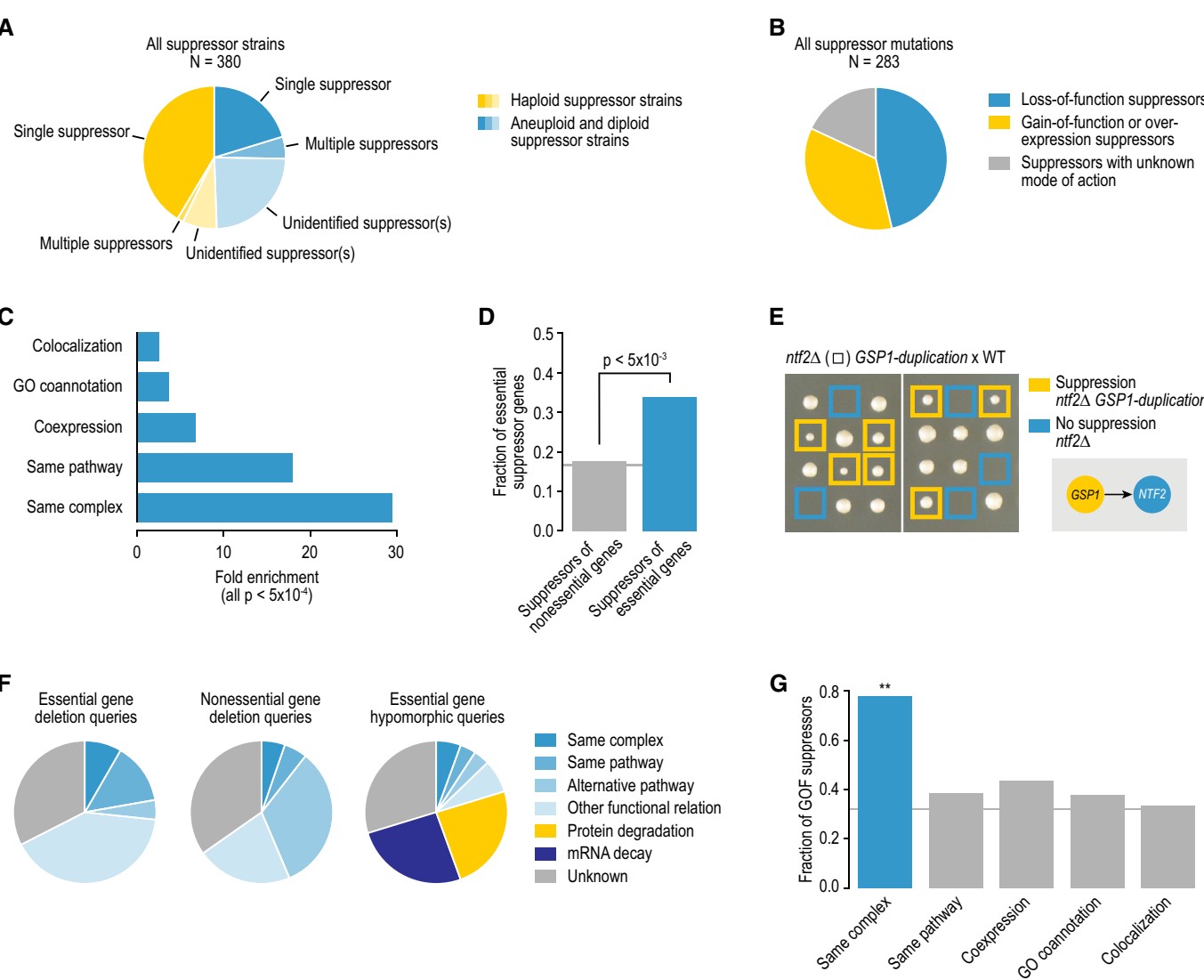

**Figure 3. Properties of essential gene bypass suppressors.**

A The fraction of all suppressor strains in which we identified one single suppressor, multiple co-occurring suppressors, or in which we were unable to identify a suppressor gene, divided by the genome content of the strain.

B The fraction of all suppressor mutations that involve loss-of-function, gain-of-function, or unknown modes of action.

C Fold enrichment for colocalization, GO coannotation, coexpression, same pathway membership, and same complex membership for gene pairs involved in a suppression interaction.

D The fraction of unique suppressor mutations that map to an essential gene, for the suppressors of essential gene deletion mutants identified in this study or suppressors of nonessential gene deletion mutants identified previously (Van Leeuwen *et al*, 2016).

E An example of a bypass suppressor of an essential gene deletion mutant. Tetrad dissection analysis of a strain heterozygous for a *ntf2*Δ deletion allele and a *GSP1-duplication* allele. Blue squares highlight the lack of colony growth associated with *nft2*Δ single-mutant cells. Yellow squares highlight colony growth of *nft2*Δ *GSP1-duplication* double mutants. WT, wild type.

F Distribution of suppression interactions across different mechanistic suppression classes, for the suppressors of essential gene deletion mutants identified in this study or suppressors of nonessential gene deletion mutants and essential gene hypomorphic alleles identified previously (Van Leeuwen *et al*, 2016).

G The fraction of gain-of-function (GOF) suppressor mutations for suppressor interactions showing different types of functional connection between the suppressor and the query gene.

Data information: Statistical significance (panels C, D, and G) was determined using Fisher's exact test, **P < 0.005. Gray lines indicate background rates of gene essentiality (D) and GOF suppression (G).

Leeuwen *et al*, 2016). Using this classification system, we found that 68% of essential gene bypass suppression interactions could be explained by a functional relationship between the suppressor and query genes, such as shared complex or pathway membership or

annotation to the same biological process (Fig 3F). This fraction of functionally related pairs is comparable to that seen for suppressors of nonessential gene deletion queries (65%) (Van Leeuwen *et al*, 2016), but significantly higher than that of suppression interactions

involving hypomorphic alleles of essential query genes (20%, Fig 3F, $P < 0.0005$ Fisher's exact test) (Van Leeuwen et al, 2016). The suppression of essential gene hypomorphic queries frequently involves mRNA or protein degradation pathways, which ultimately leads to increased activity of the partial loss-of-function allele (Van Leeuwen et al, 2016). In addition, the fraction of bypass suppressor and essential query gene deletion pairs encoding members of the same complex or pathway (~ 20%) was double that of suppressors of nonessential gene deletion mutants (~ 10%, Fig 3F) (Van Leeuwen et al, 2016).

When considering only bypass suppressor genes that encode members of the same complex as the corresponding essential query gene, ~ 80% of the suppressor mutations were associated with a gain-of-function phenotype, significantly higher than the ~ 30% gain-of-function mutations observed for all bypass suppressors (Fig 3G, Dataset EV2). Gain-of-function mutations in a gene encoding a component of the same complex as the query gene may restore complex function in the absence of the query, either by stabilizing a multimeric complex or by making the function of the query subunit obsolete (Van Leeuwen et al, 2017). For example, in three cases suppression occurred by amplification of, or a gain-of-function mutation in, a paralog of the dispensable essential gene, which is significantly more frequent than would be expected by chance (Appendix Fig S2B, $P < 0.0005$ Fisher's exact test). Only in two cases did loss of a complex member suppress the lethality of losing another subunit of the same complex: (i) the suppression of a deletion allele of RCL1, which encodes a preribosome processome complex component by loss-of-function mutations in CMS1, a highly conserved, nonessential, and relatively uncharacterized gene in the same complex; and (ii) the suppression of deletion of CDC11, which encodes an essential component of the septin complex, by loss-of-function mutations in the nonessential septin gene, SHS1. In the latter case, the interaction does not technically occur within the same complex, since Cdc11 and Shs1 occupy terminal positions in different septin hetero-octamers; Cdc11 octamers polymerize into linear filaments, whereas Shs1 octamers form more elaborate structures (Garcia et al, 2011). In the absence of CDC11, SHS1 expression becomes toxic due to the absence of linear filaments, whereas in a cdc11Δ shs1Δ double mutant, septin hexamers can still polymerize to form linear filaments (McMurray et al, 2011).

Thus, bypass suppressors of essential gene deletion mutants share several properties with suppressors of nonessential gene deletion mutants, such as a strong functional connection between the query and the suppressor gene. However, essential gene bypass suppressors more frequently involve gain-of-function mutations in essential suppressor genes or in genes encoding members of the same complex as the query gene.

### Most dispensable essential genes can only be suppressed by a single genetic mechanism

The isolation of multiple independent suppressors for most essential query genes allowed us to investigate how many different suppression mechanisms exist for a particular query gene. We focused on the 50 query genes for which we had isolated multiple independent suppressor strains, each of which carried a single suppressor mutation. In total, 20 (40%) of the query genes were suppressed by mutations in one common suppressor gene, whereas for another 30

query genes, we identified two or more different suppressor genes (Fig 4A, Dataset EV2). We note that for query genes with multiple TS alleles, the specific TS allele had no effect on the identified bypass suppressor (Appendix Fig S2C). This is expected, since our approach demands suppression of an essential gene deletion, so suppressors specific to a particular point mutation will not be identified.

We examined the number of newly identified suppressor genes for each independent suppressor isolation event and fitted a logarithmic model to the data (Fig 4B). This analysis suggests that we identified ~ 65–70% of all possible suppressor genes for the set of tested query genes. Isolating additional suppressor strains will thus likely yield more suppressor genes, although the chance of identifying a novel suppressor gene decreases for each additional suppressor isolate (Fig 4B). Moreover, when multiple suppressor genes were identified for a query gene, they were often coexpressed or encoded members of the same pathway or complex (Fig 4A, 13/30 cases). This result suggests that despite the isolation of multiple suppressor genes, there are only a few fundamental ways of rewiring biological processes or pathways through genome alteration such that deletion of an essential gene can be suppressed. For example, functionally connected suppressor genes were observed for the suppression of the lethality associated with loss of TOR complex 2, which activates a phosphorylation cascade that induces sphingolipid biosynthesis. Mutations in any of the members of this signaling pathway bypassed the essentiality of TOR complex 2 subunits by reactivating part of the signaling cascade and thereby restoring sphingolipid biosynthesis (Fig 4C).

In cases where the suppressor genes had no known functional connection among each other, the corresponding query genes tended to be more pleiotropic, with multifunctional roles (Fig 4D). Thus, although in general there are only a few routes to suppression, multiple suppression mechanisms may exist for multifunctional query genes.

### Complex suppression interactions

In 24 strains (corresponding to seven query genes), we observed the co-occurrence of suppressor mutations in two genes at the same time (Fig 3A), which were often functionally related ($P < 0.005$, permutation test). For example, the lethality associated with deletion of CDC25, which encodes the guanine nucleotide exchange factor that activates Ras2 activity, can be bypassed by the combination of a loss-of-function mutation in IRA2, which encodes a GTPase-activating protein (GAP) that negatively regulates Ras2 activity, and a specific three-base pair deletion in RAS2 that removes the highly conserved glycine residue G20 (Fig 4E, Dataset EV2) (Broek et al, 1987). Mutations in the corresponding residue in human Ras isoforms (G13) frequently drive cancer formation and lead to decreased GAP-mediated GTP hydrolysis and thus a gain-of-function phenotype of Ras (Hobbs et al, 2016).

A cdc25Δ allele could also be suppressed by the co-occurrence of loss-of-function mutations in IRA2 and its paralog, IRA1 (Fig 4E and F, Dataset EV2). In this case, detailed tetrad analysis revealed that mutations in IRA1 alone were sufficient to bypass cdc25Δ lethality, but an additional mutation in IRA2 leads to an increase in fitness of the original bypass suppressor strain (Fig 4F). The order in which the suppressor mutations occurred is likely important, as an ira2-

*T596P* mutation alone could not suppress *cdc25Δ* lethality (Fig 4F). Similarly, for 4 additional query genes (*NUP116, NUP57, SCC4,* and *SLN1*) for which we observed co-occurrence of two suppressor genes, suppressor strains carrying mutations in only one of the suppressor genes were obtained as well, suggesting that a single suppressor event is sufficient to suppress the lethality, but the combination of both mutations can increase the fitness of the query strain (Dataset EV2).

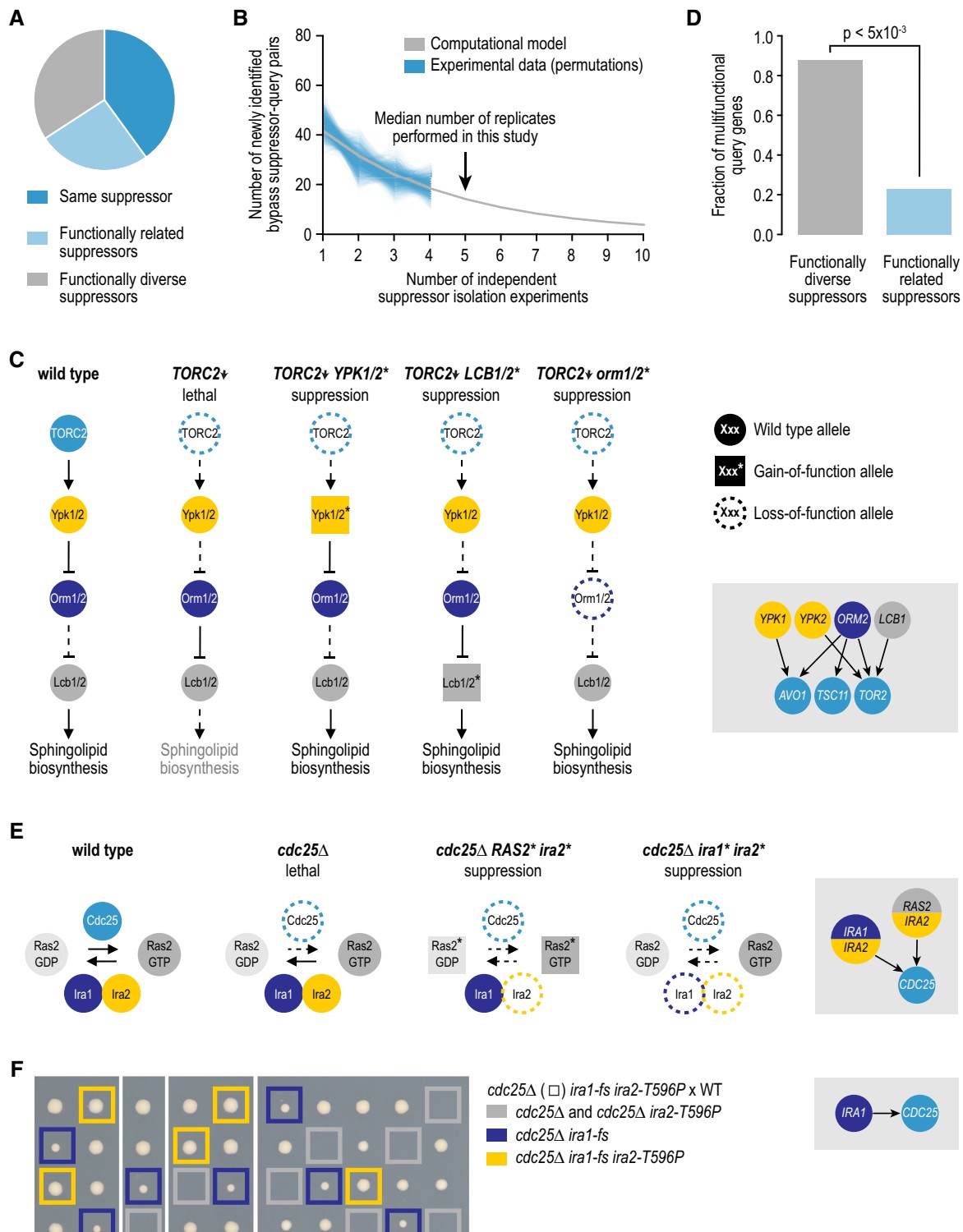

**Figure 4.**

◀

**Figure 4. Most dispensable essential genes are suppressed by a single genetic mechanism.**

A   The fraction of query genes for which suppressor mutations were identified in multiple independent suppressor strains that were suppressed by mutations within the same suppressor gene, within multiple functionally related suppressor genes, or within multiple unrelated suppressor genes.

B   The number of newly identified query–suppressor gene pairs is plotted against the number of independent suppressor isolation experiments. Shown are the 1,000 random permutations of the experimental results and a computational model that was fit to these data. Note that we used at most four independent suppressor isolation experiments per query gene for the random permutations, while a median of five experiments was performed.

C   An example of multiple suppressor genes within a pathway that can each individually suppress the same query gene mutant.

D   The fraction of query genes that are considered to be multifunctional (assigned to two or more biological processes), for queries for which multiple suppressor genes have been identified that can individually suppress the query mutant. Query genes are split into those that are suppressed by suppressor genes that are functionally diverse and those that are suppressed by functionally related suppressor genes. Significance was determined using Fisher's exact test.

E, F   (E) Examples of complex suppression interactions, in which two suppressor genes are mutated. (F) Tetrad dissection analysis of a strain heterozygous for *cdc25Δ*, *ira1-fs* (fs, frameshift), and *ira2-T596P* mutant alleles. Gray squares highlight the lack of colony growth associated with *cdc25Δ* or *cdc25Δ ira2-T596P* double mutants. Blue squares highlight the colony growth of *cdc25Δ ira1-fs* double mutants. Yellow squares highlight the colony growth associated with *cdc25Δ ira1-fs ira2-T596P* triple-mutant cells. WT, wild type.

For two query genes, *YIP1* and *YRA1*, each of their isolated bypass suppressor strains carried mutations in two independent suppressor genes simultaneously, suggesting that mutation of both genes could be required for the suppression phenotype (Dataset EV2). The lethality of a *YIP1* deletion allele was suppressed by two gain-of-function mutations, one in *YIP4* and one in *YIP5*, which encode poorly characterized members of the *YIP1* family of membrane proteins that interact with Rab GTPases to regulate membrane trafficking (Calero *et al*, 2002). The lethality associated with deletion of *YRA1*, which encodes a protein required for the export of polyadenylated mRNA from the nucleus, was suppressed by simultaneously increasing the copy number of both the *YRA1*-paralog *YRA2* and the gene encoding mRNA export factor Mex67.

To summarize, in cases where multiple suppressor mutations co-occur in a suppressor strain, either both mutations may be required for the bypass suppression phenotype, or one suppressor mutation may act as a bypass suppressor and the second mutation further improves the fitness of the suppressor strain.

**Suppression by aneuploidies and gene duplication**

Out of the 380 suppressor strains that we sequenced, 188 (49%) carried an extra copy of one or more chromosomes (Figs 3A and 5A, and Datasets EV2 and EV7). Out of the 188 strains with altered DNA content, 116 had acquired chromosome amplifications, whereas 72 strains had undergone whole-genome duplication, often accompanied by additional chromosome gains or losses (Fig 5A). In the majority of these cases (76%), whole-genome duplication appeared to be the consequence of a defect in chromosome segregation or cell division caused by the query mutation (Yu *et al*, 2006). For example, all three bypass suppressor strains of *INO80*, which encodes a member of the INO80 chromatin remodeling complex involved in the regulation of chromosome segregation (Chambers *et al*, 2012), were diploidized. In this case, suppression occurred via homozygous loss-of-function mutations in histone deacetylase genes (Dataset EV2), which likely counteract the reduced histone acetylation due to histone reorganization in *ino80* mutants (Papamichos-Chronakis *et al*, 2011; Chambers *et al*, 2012). The other diploidization cases may either be spurious events as a result of the propensity of haploid *S. cerevisiae* strains to diploidize under stressful conditions (Gerstein *et al*, 2006; Harari *et al*, 2018), or identify unappreciated roles of either the query or the suppressor gene in preventing polyploidy.

The frequency at which aneuploidies or ploidy changes were found in our suppressor strains (49%) is substantially higher than the relatively low frequency (~ 1 in a million) of aneuploid strains that are normally found in cultures of wild-type laboratory yeast strains (Mulla *et al*, 2014) or the aneuploidy rate (19%) found across hundreds of natural yeast isolates (Peter *et al*, 2018). Although the aneuploidy rate differed from wild-type populations, the relative frequency of chromosome-specific aneuploidies was conserved in our dataset and negatively correlated with chromosome size (Appendix Fig S3A and B). Aneuploidies are known to lead to a fitness cost (Torres *et al*, 2007; Beach *et al*, 2017), and the average fitness of suppressor strains carrying an aneuploid chromosome was significantly lower than that of euploid suppressor strains (Appendix Fig S3C).

For 34 query genes (66% of all aneuploid strains), the same aneuploidy was recurrently identified in independent suppressor strains, but was absent in the parental strain, suggesting that it was involved in the suppression phenotype (Fig 5A, Dataset EV2). Aneuploidies that were likely involved in the suppression phenotype were less detrimental than random aneuploidies that played no role in the suppression, despite the larger size of the former category (Appendix Fig S3D and E). Although, in theory, a gain-of-function mutation in a suppressor gene could lead to the same outcome as gene overexpression, most query genes were either always suppressed by an aneuploidy in all independent suppressor isolates or always by a suppressor SNP (Appendix Fig S3F and G). One notable exception is the query gene *NUP116*, encoding a subunit of the nuclear pore complex, for which we isolated 16 independent suppressors: 15 of these carried a chromosome VIII duplication, whereas one strain was euploid but had a gain-of-function mutation in *BRL1*, which is located on chromosome VIII and encodes a nuclear envelope protein (Dataset EV2). Consistent with these findings, overexpression of *BRL1* was previously shown to rescue the lethality associated with deleting specific nuclear pore genes (Liu *et al*, 2015).

In 16 strains, suppression occurred by amplification of only a portion of a chromosome, and these variants often increased the fitness of these strains when compared to the corresponding query mutant strains carrying a fully aneuploid chromosome (Fig 5B, Appendix Fig S4A and B, Dataset EV2). The partial amplifications typically resulted from breakpoints at repetitive sequence elements, such as transposon long terminal repeats or tRNA sequences (19 out of 21 breakpoints; Appendix Fig S4A). Although most chromosomal

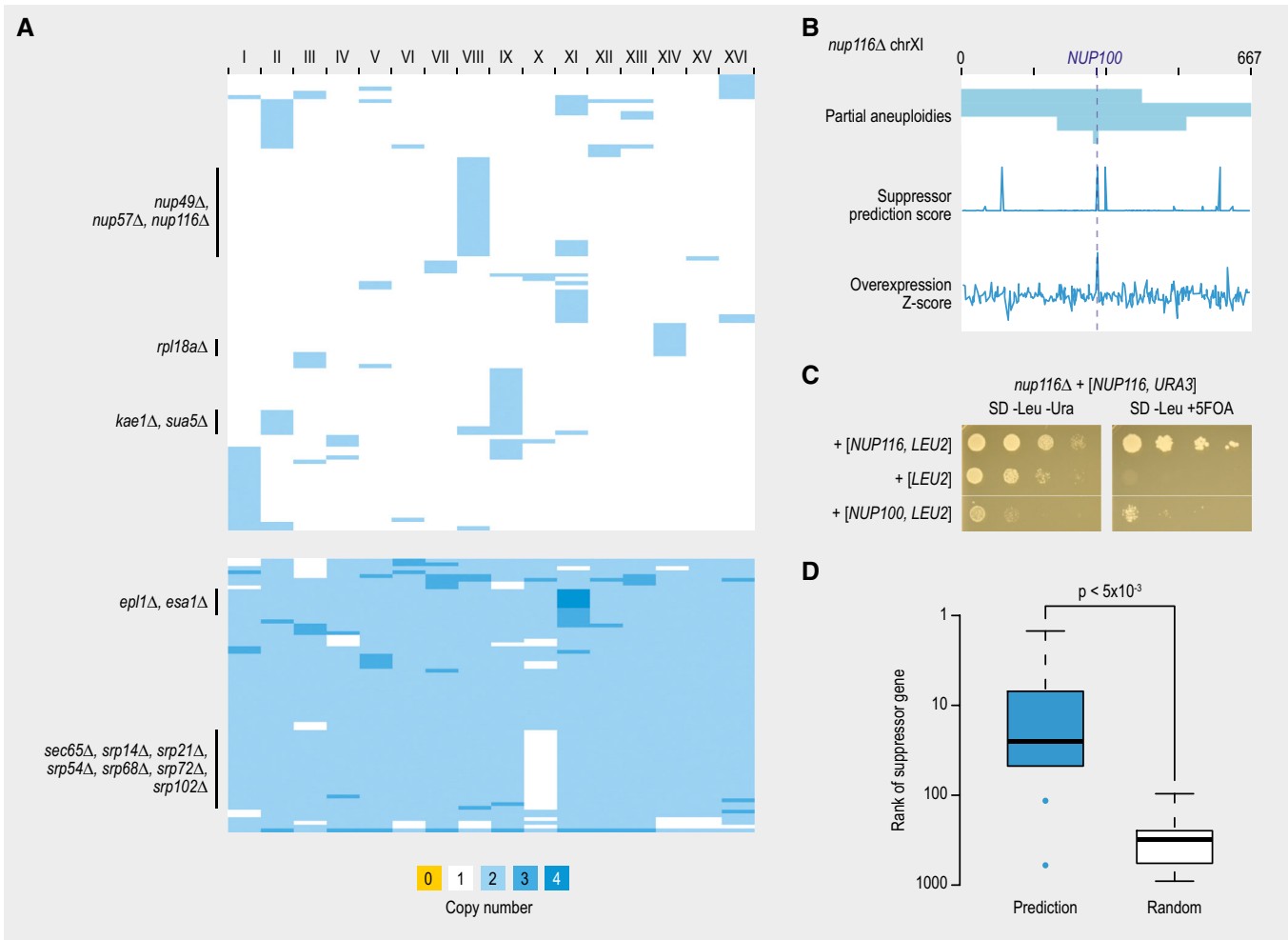

**Figure 5. Suppression by aneuploidies.**

A  Heatmap showing chromosomal copy numbers of suppressor strains that carried an aneuploidy. Each row represents a different suppressor strain. Columns correspond to each of the 16 yeast chromosomes.

B  An example of a query gene showing recurrent aneuploidies. Suppressor strains of *nup116Δ* lethality frequently show an amplification of chromosome XI. In some cases, this amplification is only partial (top). A suppressor prediction algorithm was used to predict the causal suppressor gene on chromosome XI based on functional information (middle). Overexpression of each gene on chromosome XI individually confirmed one of the predicted suppressor genes (*NUP100*) as the actual suppressor (bottom).

C  Suppression of *nup116Δ* lethality by overexpression of *NUP100*. Cultures of the indicated strains were diluted to an optical density at 600 nm of 0.1, and a series of 10-fold dilutions was spotted on agar plates and incubated at 30°C for 2–3 days.

D  Comparison of the median rank of confirmed suppressor genes (*N* = 10), either in a list of genes ranked by the likeliness of being a suppressor gene using our suppression prediction algorithm or in a random gene list. Statistical significance was determined using a Mann–Whitney *U*-test. The central bands in the box plot are the median values. Boxes represent data between the first and third quartiles. Upper and lower whiskers extend to the largest and smallest values, respectively, excluding outliers which are shown as dots. Outliers are values outside the range [Q1 − (1.5 × IQR), Q3 + (1.5 × IQR)].

fragments were duplicated, several query mutants encoding proteins involved in the transcription of rDNA were suppressed by threefold to 10-fold amplification of the ribosomal DNA locus *RDN1* (Dataset EV7, Appendix Fig S4B).

### Predicting suppressor genes on aneuploid chromosomes

An extra copy of a chromosome in a haploid cell generally leads to a twofold increase in expression of the genes on the disomic chromosome (Torres *et al*, 2007; Pavelka *et al*, 2010). However, typically, overexpression of only one or two genes is responsible for the

beneficial effect of an aneuploidy (Chen *et al*, 2012; Kaya *et al*, 2015; Liu *et al*, 2015; Linder *et al*, 2017). To identify the causal suppressor gene among the genes on the aneuploid chromosome(s), we developed a suppressor prediction algorithm that exploited the strong functional connection generally observed between suppressor and query genes (Figs 2 and 3C and F). In brief, each gene on the aneuploid chromosome was given a suppressor prediction score depending on four different measures of functional connection with the query gene: colocalization, coexpression, and shared complex or pathway membership. Those properties representing a close functional connection, such as shared complex membership, were

weighted more heavily than more distant relationships, such as colocalization (see Materials and Methods). Genes were subsequently ranked based on their suppressor prediction score. This method can be used to predict candidate suppressor genes for any query gene and aneuploid chromosome pair, but the quality of the predictions will be dependent on the availability of functional data for the query and suppressor genes. We used this suppressor prediction approach to identify candidate suppressor genes on each of the detected aneuploid chromosomes (Fig 5B, Dataset EV9).

To experimentally validate our suppressor predictions, we systematically overexpressed all genes on the disomic chromosomes individually in 53 different euploid parental query strains and tested whether the resulting overexpression mutants could survive loss of the essential query gene (Appendix Fig S4C and Dataset EV10). As a negative control, we included all cases of aneuploidies that were thought to be spurious events with no role in the suppression phenotype. All six cases in which the query gene itself was overexpressed showed up as a hit in the screens. For the 30 query genes for which the aneuploidy appeared to be a spurious event, because either a SNP suppressor event had been identified in the suppressor strains, or the aneuploidy had occurred in only one out of several independently isolated suppressor strains, we identified a unique overexpression suppressor for only one query gene (3%). Out of the 23 query genes whose suppressor strains showed reoccurring aneuploidies of the same chromosome and in which no suppressor SNPs were identified, we identified overexpression suppressors for nine query genes (39%). For instance, *nup116Δ* lethality was suppressed by increased copy number of its paralog *NUP100* (Fig 5B and C). Both genes encode highly similar nucleoporin components of the central core of the nuclear pore complex, and Nup100 may thus potentially replace Nup116 in the central core (Bailer *et al*, 1998). For the remaining 14 query genes that appeared to carry a suppressor aneuploidy but for which we did not identify an overexpression suppressor, overexpression of multiple genes simultaneously may have been involved in the suppression phenotype. For example, four suppressor strains of *TRM5*, encoding a tRNA methyltransferase, carried aneuploidies of both chromosomes I and II, suggesting that both aneuploidies may contribute to the suppression phenotype.

For the 10 query genes for which we identified an overexpression suppressor experimentally, five of the suppressor genes ranked among the top 15 of those predicted, with two suppressors ranking in the top 5 (Fig 5D, Dataset EV2). Thus, the various functional properties identified for suppressor genes (Fig 3) can narrow the search space for potential suppressor genes associated with an aneuploidy from hundreds to tens of genes.

### Conservation of bypass suppression interactions in diverse yeast strains

Because some dispensable essential genes that were characterized in other *S. cerevisiae* genetic backgrounds were not observed in our assay involving the reference background, S288c (Dataset EV3), we suspected that bypass suppression interactions could be affected by genetic background variation. To test this hypothesis, we investigated the conservation of bypass suppression interactions involving loss-of-function suppressors in three diverse *S. cerevisiae* strains isolated from different environments, including strains isolated from

a winery in Italy (FIMA_3), an oak tree in Canada (ZP_611), and a hickory tree in China (SX3), which show 0.35, 0.48, and 0.91% genetic divergence from the S288c reference strain, respectively (Fig 6A) (Peter *et al*, 2018). We tested 10–13 bypass suppression interactions per yeast strain, with 8 interactions tested in all three genetic backgrounds (Dataset EV11). In FIMA_3, the strain that is most closely related to S288c, the lethality of deleting the tested query genes was suppressed by deletion of the suppressor gene identified in S288c in the majority (8/10) of the cases (Fig 6B, Dataset EV11). One of the tested query genes was not essential in this genetic background, while one other query was essential but not suppressed by deletion of the suppressor gene identified in S288c (Fig 6B, Dataset EV11). With increased genetic divergence, the fraction of conserved suppression interactions decreased, and the fraction of query genes that was nonessential in the strain background rapidly increased (Fig 6B, Dataset EV11). The loss of query gene essentiality in the distantly related strains suggests that one or more suppressor modifiers are present in these genetic backgrounds. The cases in which the tested dispensable query gene was essential but did not show the corresponding bypass suppression also increased with genetic divergence, but remained relatively rare (Fig 6B). These observations suggest that genetic background diversity has a significant impact on the specific set of dispensable essential genes within a genome.

### Dispensable essential genes show distinct evolutionary signatures

To investigate potential differences in evolutionary pressures between dispensable and indispensable essential genes, we used available data from model organism databases and systematic gene perturbation studies (Dowell *et al*, 2010; Blomen *et al*, 2015; Lock *et al*, 2018; Segal *et al*, 2018; Harris *et al*, 2020) to compare gene essentiality across yeasts and other species (Materials and Methods). Dispensable essential genes were in general more likely to be nonessential in another *S. cerevisiae* strain (Appendix Fig S5A) and other yeast species (Fig 6C and D, Appendix Fig S5B) than indispensable essential genes. Dispensable essential genes were also less conserved in more distant species; they were more frequently absent, duplicated, or nonessential than indispensable essential genes in worms, and they were depleted among a set of 1750 essential genes shared by the highly related human cell lines, KBM7 and HAP1 (Fig 6C and D, Appendix Fig S5B). Notably, dispensable essential genes for which either the fitness could be restored to wild-type levels or that could be suppressed by multiple suppressor genes (Dataset EV2) were more frequently absent or nonessential in other species than other dispensable genes (Appendix Fig S5C and D). Thus, dispensable essential genes are less conserved than other essential genes, suggesting that bypass suppressors isolated in the laboratory might reflect suppression events that occur during evolution.

To investigate whether human genes that are essential in only a subset of cell lines, often referred to as context-dependent or selective essential genes, show comparable characteristics to our dispensable essential yeast genes, we examined gene essentiality data obtained from genome-scale CRISPR-Cas9 genetic perturbation reagents for 18,333 human genes across 739 cell lines, from the Cancer Dependency Map (DepMap) project (Meyers *et al*, 2017). In

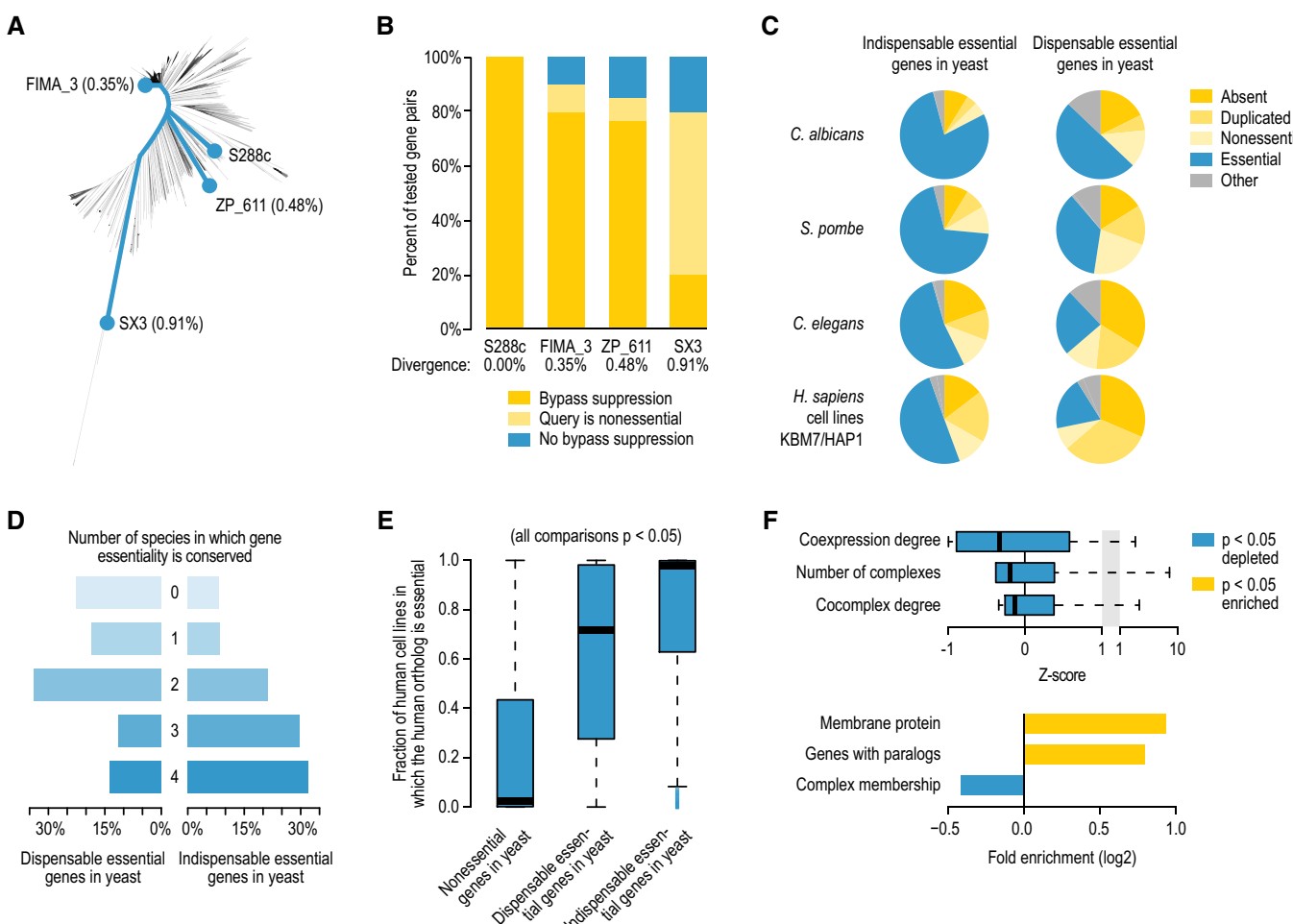

**Figure 6. Evolutionary properties of dispensable essential genes.**

A  Phylogenetic tree of 1,011 *Saccharomyces cerevisiae* strains, highlighting the laboratory strain S288c, and three strains isolated from a winery (FIMA_3) and from the bark of oak (ZP_611) and hickory (SX3) trees (Peter *et al*, 2018). Percentages indicate the genetic divergence from S288c.

B  The fraction of bypass suppression interactions that are conserved in three different *S. cerevisiae* strains.

C  The fraction of dispensable and indispensable essential genes in *S. cerevisiae* that have orthologs that are absent, duplicated, nonessential, or essential in *Candida albicans*, *Schizosaccharomyces pombe*, *Caenorhabditis elegans*, or *Homo sapiens* cell lines KBM7/HAP1.

D  The fraction of dispensable or indispensable essential query genes that are not essential in 0, 1, 2, 3, or 4 of the species indicated in (C).

E  Distribution of the fraction of human cell lines in which 1-to-1 orthologs of nonessential, dispensable essential, and indispensable essential yeast genes are essential, using a set of 739 human cell lines (Meyers *et al*, 2017). Statistical significance was determined using Mann–Whitney *U*-tests. The central bands in the box plot are the median values. Boxes represent data between the first and third quartiles. Upper and lower whiskers extend to the largest and smallest values, respectively, excluding outliers which are shown as dots. Outliers are values outside the range [Q1 − (1.5 × IQR), Q3 + (1.5 × IQR)].

F  Enrichment of context-dependent essential human genes for various gene- and protein-level properties. Fisher's exact or Mann–Whitney *U*-tests were performed to determine statistical significance of the results. Box plot: The central band is the median value. Boxes represent data between the first and third quartiles. Upper and lower whiskers extend to the largest and smallest values, respectively.

addition to orthologs of dispensable essential yeast genes being more frequently nonessential in human cell lines, we found that they were also often essential in only a subset of cell lines, indicating that they are context-dependent essential human genes (Fig 6E, Appendix Fig S6A and B). Similar to dispensable essential yeast genes, context-dependent essential human genes had significantly more paralogous genes and a lower coexpression degree when compared to genes that were essential in the majority of cell lines (Figs 1D and 6F, and Appendix Fig S6C and D). Moreover, as we observed in yeast, these context-dependent essential human genes were depleted for genes encoding members of protein complexes,

particularly large complexes, and were frequently membrane-associated (Figs 1D and 6F, Appendix Fig S6C and D). Finally, context-dependent essential human genes were more frequently absent in other species than indispensable essential human genes (Appendix Fig S6E). Thus, essential gene dispensability and its characteristics appear to be conserved from yeast to human.

## Predicting gene dispensability

Given the distinct functional and evolutionary properties of dispensable essential genes compared to other essential genes (Figs 1 and

6), we developed a model that uses these signatures to identify dispensable essential yeast genes (see Materials and Methods). In brief, we used a set of diverse features, including ortholog essentiality in various species (Fig 6), gene function (Fig 1C), and various gene and protein properties such as coexpression degree and complex membership (Fig 1D), to train a random forest classifier. We evaluated the performance of our model by excluding a subset of our data from the training set and by using dispensable essential gene sets identified in other studies (Liu *et al*, 2015; Van Leeuwen *et al*, 2016) but not tested in our experiments. Our method showed similar predictive power in all datasets yielding an average area under the receiver operating characteristic curve of 0.76 (Fig 7A, Appendix Fig S7).

Next, we applied our prediction model to the 329 essential genes that were not present in our query strain collection and thus were not tested for bypass suppression in our experiments (Dataset EV12). This analysis identified 82 essential genes for which the prediction score of the gene being dispensable was above 0.5. We ranked the 329 essential genes by their dispensability prediction score and selected the 13 highest and 15 lowest scoring genes for which TS alleles were available for experimental validation. For each of these 28 query genes, we constructed query strains and tested ~ 50 million cells, involving two independent experiments, for the occurrence of spontaneous bypass suppressor mutations (Fig 7B, Dataset EV12). For seven (54%) of our predicted dispensable genes, we could indeed isolate viable suppressor strains that lacked the essential gene. Given the false-negative rate associated with two experimental replicates (Fig 1B), an additional ~ 1–2 of the tested query genes are likely dispensable essential. Importantly, we failed to isolate any bypass suppressors for any of the 15 tested genes that we predicted to be indispensable (*P* < 0.005, Fisher's exact test, Fig 7B, Dataset EV12).

We sequenced the genomes of the obtained bypass suppressor strains to determine the identity of the suppressors. In three cases, the bypass suppression involved an aneuploidy, and in two cases, we identified a point mutation within a single suppressor gene, whereas in another two cases, the suppressor remained unidentified (Dataset EV12). All identified suppressor genes (Dataset EV12) showed a functional connection to their corresponding query genes, which is consistent with the general trends observed in our large-scale study (Figs 2 and 3). For example, one of the query genes, *SSY5*, which encodes an essential member of the Ssy1-Ptr3-Ssy5 amino acid sensor, was suppressed by a deleterious mutation in the phosphatase Sit4 (Dataset EV12). Upon the detection of amino acids, Ssy5 activates the transcription factor Spt1, which induces the transcription of amino acid permease genes. Sit4 negatively regulates Spt1 (Shin *et al*, 2009), suggesting that the Sit4 mutation may suppress the lethality of a *ssy5Δ* deletion mutant via increased Spt1 activity and thus improved amino acid uptake in the absence of a functional amino acid sensor.

Finally, we defined a list of core essential yeast genes that were either found to be indispensable in our experiments or predicted to be indispensable using a stringent cutoff (see Materials and Methods). This resulted in a list of 805 essential genes that appear to be absolutely required for cell viability in yeast (Dataset EV13). Thus, based on the functional and evolutionary properties that distinguish dispensable from core essential genes, we were able to predict dispensable essential genes among the genes that had not yet been

experimentally assessed and to define a core set of essential yeast genes.

## Discussion

We systematically assessed the genetic context dependency of the essentiality of 728 budding yeast genes and found that ~ 17% of the tested essential genes (124 genes) were dispensable and subject to bypass suppression in our assay. There was no previous evidence for the dispensability of about half (60) of these genes, and their identification highlighted biological functions, protein complexes, and gene properties that can make an essential cellular component nonessential in a specific genetic context (Fig 1). A previous study estimated the percentage of budding yeast dispensable essential genes to be ~ 9% (Liu *et al*, 2015), but in this analysis suppressors were scored following germination of a single deletion mutant spore, which means that about a million-fold fewer cells were examined per query gene. Nearly all (95%) of the identified suppressor strains using the spore-based approach showed substantial ploidy changes. Even though most aneuploidies come at a fitness cost (Torres *et al*, 2007; Beach *et al*, 2017) (Appendix Fig S3C), changes in chromosome number may be the only available route for suppression of severe growth defects within relatively small populations of cells, as mutation rates are generally low and thus specific suppressor point mutations are unlikely to arise within a single spore or a relatively small colony (Lang & Murray, 2008). By contrast, a substantial fraction of suppressor strains (51%, Fig 3A) we identified in our study are haploid and often carry a SNP suppressor event. Most query genes were suppressed either always by a SNP or always by an aneuploidy, in independent suppressor isolates (Appendix Fig S3F and G), suggesting that there are query gene-specific genome rewiring mechanisms and explaining the observed differences in frequency of dispensability.

A limited survey of ~ 10% of essential genes in the fission yeast *Schizosaccharomyces pombe* found bypass suppressors for ~ 27% of the tested essential gene deletion alleles (Li *et al*, 2019), which is significantly higher than the ~ 17% gene dispensability we find in *S. cerevisiae*. The fission yeast study used chemical mutagenesis, transposon insertions, and artificial gene overexpression to identify potential suppressor genes and may thus have identified suppressor mechanisms that are difficult to achieve by spontaneous genomic alterations. While rare genomic point mutation suppressors showed a strong functional connection to the deleted essential gene, as we observed in *S. cerevisiae*, bypass suppression by gene overexpression frequently involved functionally unrelated suppressor genes that may affect cellular homeostasis (Appendix Fig S8) (Li *et al*, 2019). The observed difference in dispensability frequency may also result from the higher fraction of essential genes in fission yeast compared to budding yeast (26 versus 17%, respectively) (Giaever *et al*, 2002; Kim *et al*, 2010). Indeed, many of the identified *S. pombe* dispensable essential genes had a role in mitochondrial respiration, which is essential for viability in *S. pombe* but not in several other yeasts, including *S. cerevisiae* (Li *et al*, 2019).

For 329 essential genes, no query strains were present in our collection, and therefore for an additional ~ 55–60 essential genes (~ 17% of 329), viability upon gene loss may be dependent on the genetic context. Using the functional and evolutionary properties

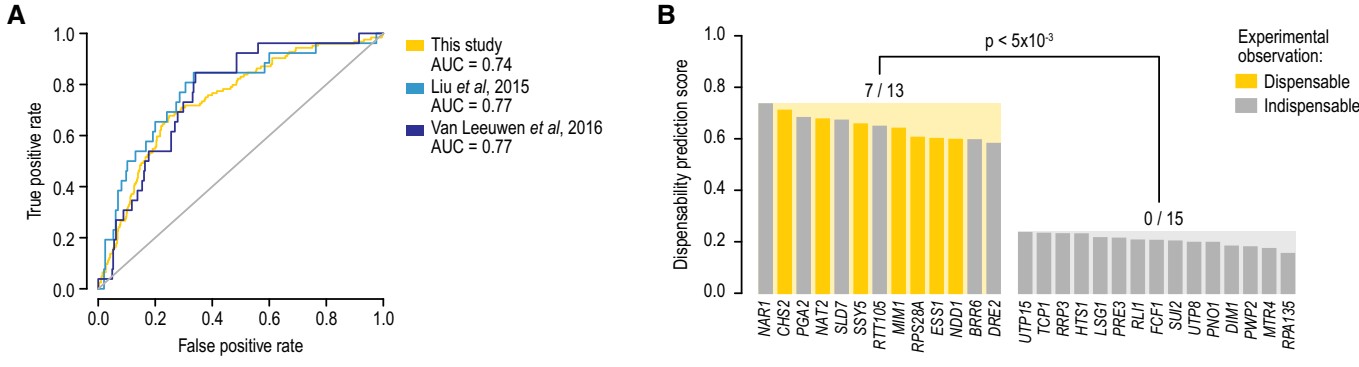

**Figure 7. Predicting essential gene dispensability.**

A   A dispensable essential gene prediction model was developed based on the distinct functional and evolutionary properties of dispensable essential genes compared to other essential genes, and the performance of the model was evaluated. The true-positive rate was plotted against the false-positive rate of the dispensable essential gene prediction model. True-positive dispensable essential genes were defined either by excluding a subset of dispensable essential genes found in this study from the training set ("this study") or by using dispensable essential gene sets identified in other studies but not tested in our experiment (Liu *et al*, 2015; Van Leeuwen *et al*, 2016).

B   For 13 essential genes that were predicted to be dispensable and 15 genes predicted to be indispensable, we experimentally tested whether we could identify viable suppressor strains that lacked an essential gene. In each case, we tested for bypass suppression in two independent assays. Experimentally observed dispensable essential genes are highlighted in yellow. The *P* value indicates the statistical significance of the difference between the number of observed dispensable essential genes between the gene sets predicted to be dispensable or indispensable (Fisher's exact test).

that we defined for dispensable essential genes (Figs 1 and 6), we generated a list with likely candidates for dispensability (Datasets EV12 and EV13). In addition, we combined our experimental data and our computational predictions to define a list of 805 core essential genes that are likely required for viability in yeast regardless of the genetic context (Dataset EV13). Some of these essential genes could be dispensable in different environments or in the presence of more sophisticated genomic rewiring that cannot be easily achieved by spontaneous mutation, such as specific rare missense variants, or the simultaneous mutation of multiple suppressor genes. Although we found a few examples of multiple suppressor mutations within one strain (Fig 4E and F), identifying these cases systematically would require substantially increasing mutation rates or the number of cells we use in our selection assay.

For 121 suppressor strains (32%), we were unable to identify the suppressor gene (Fig 3A). The majority of these strains (65) had either undergone whole-genome duplication, sometimes accompanied by loss of a specific chromosome, or carried complex combinations of multiple aneuploidies, suggesting that genes on multiple chromosomes may be responsible for the suppression phenotype. In either of these cases, systematically overexpressing single genes individually (Fig 5, Appendix Fig S4C) would not be expected to identify the causal suppressor gene(s). In addition, the 121 strains with unknown suppressor genes included 30 haploid suppressor strains in which no suppressor SNPs could be identified (Fig 3A). One possibility is that in these cases, the suppression phenotype is caused by structural variants, which are difficult to identify with the short-read sequencing approaches that we used.

In cases where the human ortholog of a dispensable essential yeast gene has been associated with disease, the bypass suppressors may highlight genes that are associated with a protective effect against the disease. For example, mutations within a functionally relevant interface of *TIF6* (human *EIF6*), encoding a pre-60S

ribosome nucleolar shuttling factor, bypass the fitness defect associated with deletion of *RIA1* (*EFL1*) (Dataset EV2). *EFL1* encodes a cytoplasmic GTPase that acts in concert with Sdo1, the yeast homolog of the Shwachman–Diamond syndrome protein, to promote release of Tif6 from 60S ribosomal subunits during their final maturation (Menne *et al*, 2007). Loss of one copy of the human *EIF6* gene is a recurrent finding in bone marrow cells of Shwachman–Diamond syndrome patients and is associated with a relatively benign clinical course (Pressato *et al*, 2012; Valli *et al*, 2019), suggesting that this bypass suppressor mechanism is conserved from yeast to humans (Weis *et al*, 2015; Tan *et al*, 2019), and supporting a rationale for the development of small-molecule eIF6 suppressor mimics for the treatment of Shwachman–Diamond syndrome.

We showed that dispensable essential yeast genes are often nonessential in other *S. cerevisiae* backgrounds (Fig 6B, Appendix Fig S5A), suggesting that dispensable essentiality and conditional essentiality (i.e., differences in gene essentiality between genetic backgrounds) are closely related and that bypass suppressors isolated in the laboratory might reflect suppression events that occur during evolution. However, as most of our bypass suppressor strains have a fitness defect compared to wild-type strains (Dataset EV2), we suspect that multiple suppression variants may be present in the nonreference genetic backgrounds to achieve wild-type fitness in the absence of the conditional essential gene. Indeed, we have previously shown that complex networks of genetic modifiers often underlie differences in gene essentiality between two yeast strains (Hou *et al*, 2019), and here, we found that multiple suppressors can combine to increase the fitness of the suppressor strain.

We found that human orthologs of dispensable essential yeast genes often show variable essentiality within the context of the different human cell lines queried in the DepMap project (Fig 6E, Appendix Fig S6). As the average cell line shares ~ 75% of its

essential genes with other cell lines (Hart *et al*, 2015), ∼ 25% of essential genes in any given cell line could be classified as dispensable essential, suggesting that differences in gene essentiality, or context-specific essential genes, may also be relatively common in the genomes of more complex cells. We showed that human genes that were essential in only a subset of cell lines displayed similar gene and protein properties compared to dispensable essential yeast genes (Figs 1D and 6F, Appendix Fig S6), indicating that the main characteristics that determine whether an essential cellular component is nonessential in some genetic backgrounds are conserved across species. Understanding gene dispensability and the underlying genetic rewiring may provide insight on how genetic variance accumulates during evolution and affects genetic traits, including human disease, and may identify new drug targets for bypassing the deleterious effects associated with human disease genes (Chen *et al*, 2016b).

# Materials and Methods

## Yeast strains, plasmids, and growth assays

### Yeast strains and plasmids

All used yeast strains were isogenic to S288c. The suppressor strains are listed in Datasets EV2 and EV14. For suppressor confirmation experiments, the suppressor strains were crossed to the appropriate mutant strain of the opposite mating type from either the BY4741 deletion mutant collection (*MAT*a *xxxΔ:: kanMX4 his3Δ1 leu2Δ0 ura3Δ0 met15Δ0*; Euroscarf), the SGA nonessential gene deletion mutant collection (*MATα xxxΔ:: natMX4 can1Δ::STE2pr-SpHIS5 lyp1Δ his3Δ1 leu2Δ0 ura3Δ0 met15Δ0*) (Costanzo *et al*, 2010), or the corresponding *MAT*a and *MATα* collections of DAmP or TS mutants of essential genes (Costanzo *et al*, 2016). For the plasmid complementation confirmation assays, plasmids from either the MoBY-ORF 2.0 (native promoter, 2μ, *LEU2*, *kanMX4*) (Magtanong *et al*, 2011) or the FLEX (*GAL1* promoter, *CEN/ARS*, *URA3*) (Hu *et al*, 2007) collection were used. All other strains and plasmids used in this study are listed in Dataset EV14.

### Growth, fitness, and spot dilution assays

Yeast strains were grown using standard rich (YPD) or minimal (SD) media. To determine the fitness of the suppressor strains (Dataset EV2), all suppressor strains and 104 wild-type controls (Y8835, Dataset EV14) were arrayed in duplicate in random positions across three 384-density agar plates. A border (the first and last columns and rows) of wild-type strains was added. The three 384-density plates were pinned in quadruplicate onto three agar plates to generate an array consisting of 1,536 yeast colonies per plate, on which each suppressor strain was present eight times (in quadruplicate at two positions). Four copies of this array were made: two on SDall media and two on YPD, one of each was incubated at 26°C and one at 30°C, and plates were imaged after 2 days. The images were processed using image processing software that measures colony area in terms of pixels (Wagih & Parts, 2014). We averaged the colony sizes for all eight colonies per suppressor strain for each media and temperature combination. Border strain values were removed, and suppressor strain colony sizes were normalized

against the average Y8835 colony size for each media and temperature combination. Because the differences in relative fitness between the two types of media and two temperatures were minimal, the normalized fitness values were averaged across the four conditions to yield a final fitness score. Fitness scores for strains that did not pin properly due to a rough colony morphology were manually removed.

### TS-allele-on-plasmid collection construction

To construct a collection of haploid strains, each carrying a deletion allele of an essential gene, but viable because of a TS mutant allele of the same essential gene on plasmid, we first switched the *kanMX4* cassette of essential gene mutants of the BY4743 heterozygous deletion mutant collection (*MAT*a/α *xxxΔ::kanMX4/XXX his3Δ1/his3Δ1 leu2Δ0/leu2Δ0 ura3Δ0/ura3Δ0 met15Δ0/MET15 lys2Δ0/LYS2*; Euroscarf), either to *Kluyveromyces lactis LEU2* (*KlLEU2*) followed by the C-terminal half of *natMX4* or to a nourseothricin resistance cassette followed by the C-terminal half of *kanMX4*. The C-terminal halves of *natMX4* or *kanMX4* were present to allow for testing of integration of the TS allele into the genome, which would reconstitute the complete selection cassettes (see below). For the marker switch to *KlLEU2*, we transformed the BY4743 heterozygous deletion mutants with plasmid p7413 containing a fragment of *kanR* (base pair 52–198), directly followed by the *KlLEU2* gene (without start codon) and its native terminator, and by the C-terminal half of *natMX4*, including the *Ashbya gossypii (Ag) TEF1* terminator. Initial transformants were selected using the *URA3* marker present on the plasmid, after which SD-Leu was used to select for integration events. Note that recombination occurs at the *kanR* fragment and at the *AgTEF1* terminator, leaving a small bit (198 bp) of *kanR* in front of the *KlLEU2* gene. Similarly, for the marker switch to a nourseothricin resistance cassette, we used plasmid p7412, which contains a fragment of *kanR* (base pair 52–198), directly followed by the *nat1* (*nrsR*) gene without start codon, the *AgPGK1* terminator, and the C-terminal half of *kanMX4*, including the *AgTEF1* terminator.

Next, we PCR-amplified TS alleles from available TS strains (Costanzo *et al*, 2016), thereby including regions of homology to either plasmid p7417, p7416, or p7414 (Appendix Fig S1A, Dataset EV14). These plasmids carry the counterselectable marker *URA3*, a haploid selection cassette (the promoter of either *AgSTE3* or *AgMFA2* driving the hygromycin resistance gene *hph*, followed by the terminator of either *AgCYC1* or *AgMFA2*), and directly downstream of the TS allele insertion site the N-terminal half of either *nat1* or *kanR* driven either by the *NMT1* promoter of *S. pombe* or by a synthetic promoter (de Boer *et al*, 2014). The PCR product and linearized plasmid were cotransformed into one of the marker-switched diploid yeast strains that were heterozygous for a deletion allele of the corresponding essential gene (Appendix Fig S1A). The resulting diploid strains carrying an assembled plasmid were sporulated, and haploid progeny carrying the deletion allele of the essential gene and the TS allele on plasmid were selected using the haploid selection cassette present on the plasmid. The final (simplified) genotypes were *MATα xxxΔ:: KlLEU2_natR(Cterm) his3Δ1 leu2Δ0 ura3Δ0 [xxx-ts_natR(Nterm), AgSTE3pr-hphR, URA3]* and *MAT*a *xxxΔ::natR_kanR(Cterm) his3Δ1 leu2Δ0 ura3Δ0 [xxx-ts_kanR(Nterm), AgMFA2pr-hphR, URA3]* (Dataset EV14).

### Bypass suppressor isolation

For each TS-allele-on-plasmid strain, 4–6 agar plates with ~ 25 million cells each were incubated at a range of temperatures close to the restrictive temperature of the TS allele for several days (Fig 1A). Cells from different colonies were used for each agar plate, and the 4–6 replicates were spread over independent experiments. Most cells will not be able to grow at the restrictive temperature, except for those that acquire a spontaneous suppressor mutation. When growth was observed, cells were transferred from the restrictive temperature plates onto agar plates containing 5-fluoroorotic acid (5-FOA), which is toxic to cells expressing the URA3 gene that is present on the plasmid carrying the TS allele (Boeke et al, 1984). The 5-FOA thus selected for loss of the TS allele and was therefore an assessment of whether strains could grow in the absence of the essential gene (Fig 1A). When 5-FOA-resistant colonies were obtained, loss of the plasmid was further confirmed by testing for loss of drug resistance associated with a second selectable marker that was present on the plasmid (the hygromycin resistance gene hph), and the possibility of integration of the TS allele at its endogenous locus was excluded by testing for reconstitution of a drug selection marker that was split between the C-terminus of the TS allele and the corresponding genomic deletion allele (see the previous section "TS-allele-on-plasmid collection construction"). Finally, we isolated a single 5-FOA-resistant suppressor colony per agar plate and verified absence of the TS allele by PCR analysis, using a primer internal to the essential gene, and one with homology to the region directly upstream of the TS allele on the plasmid. Datasets EV1 and EV2 contain lists of the number of times each essential gene mutant strain was independently tested for bypass suppression and the dispensable essential genes that were identified.

### Suppressor identification and confirmation

#### Synthetic genetic array mapping

Synthetic genetic array analysis was used to identify the genomic region in which the bypass suppressors were located (Jorgensen et al, 2002). In a typical SGA screen (Tong et al, 2001), a specific natMX-marked query mutation is crossed to an array of ~ 5,000 kanMX-marked deletion mutants, and in a series of subsequent pinning steps, haploid natMX- and kanMX-marked double mutants are selected. This not only generates a complete set of double mutants, but it also represents a genome-wide set of two-factor crosses, which enables us to scan the query strain genome for the presence of an unmarked extragenic suppressor locus (Jorgensen et al, 2002). When kanMX-marked deletion alleles derived from the array strains are positioned at a relatively short genetic distance from the suppressor mutation derived from the query strain, double-mutant meiotic progeny carrying the kanMX-marked deletion tend not to carry the suppressor allele. Thus, for a collinear series of ~ 20 array genes in linkage with the suppressor locus, double-mutant colonies show a reduced size (Jorgensen et al, 2002).

To be able to map the bypass suppressors by SGA, the SGA markers (can1Δ::STE2pr-SpHIS5 and lyp1Δ) that are used to select haploid cells had to be introduced into the suppressor strains. First, we transformed strain Y7091 (MATa) and Y7092 (MATα) that both carry the SGA markers, with plasmid p7415, which contains a hygromycin resistance gene under control of a MATα-specific promoter (AgSTE3-promoter, Dataset EV14). These strains were crossed to the suppressor strains of the opposite mating type, diploids were selected and sporulated, and media containing canavanine, thialysine, hygromycin B, and the appropriate selection for the essential gene deletion allele were used to isolate MATα strains carrying the essential gene deletion allele, the suppressor mutation, and the SGA markers. Note that although we are not directly selecting for the spontaneous suppressor mutation, cells carrying a deletion allele of the essential gene should be inviable in the absence of the suppressor mutation, and all selected cells should thus carry the suppressor. Finally, 5-FOA was used to remove the plasmid p7415, resulting in a collection of SGA-compatible bypass suppressor strains.

Synthetic genetic array mapping was performed on 89 suppressor strains that had a relatively mild fitness defect and did not carry aneuploidies (Datasets EV2 and EV7), because mutant strains with a severe fitness defect or aneuploidies do not make it through the SGA screening procedure. SGA analysis was performed as described previously (Baryshnikova et al, 2010a), with the exception that a smaller, condensed version of the nonessential gene deletion mutant array was used, on which each nonessential gene deletion mutant was present once, instead of four times. Potential suppressor loci were detected by visual inspection of the SGA scores (Dataset EV8).

#### Sequencing, read mapping, and SNP and indel calling

Strains were sequenced on the Illumina NextSeq 500 platform using paired-end 75-bp reads, with an average read depth of 39 across all strains. Reads were aligned to the UCSC reference sacCer3 (equivalent to SGD S288c reference genome version R64.1.1) using Bowtie2 (Langmead & Salzberg, 2012). Pileups were processed using SAMtools (Li et al, 2009) and Picard tools (http://broadinstitute.github.io/picard/). Variants were called using GATK (McKenna et al, 2010) using the following parameters: $QD = 10$, $MQ_{SNP} = 36$, $FS_{SNP} = 60$, $MQ_{indel} = 10$, $FS_{indel} = 200$ (where QD is the variant confidence divided by the unfiltered depth of nonreference samples; MQ is the root-mean-square of the mapping quality of the reads across all samples; and FS is the phred-scale-transformed P value when using Fisher's exact test to detect strand bias). The consequence of detected variants was determined using Ensembl's VEP (McLaren et al, 2016). Structural variants were detected using Manta (Chen et al, 2016c). To exclude pre-existing variants as well as systematic sequencing artifacts, variants were removed from consideration if they were present in 5 or more strains. On average, we detected 2–3 unique, nonsynonymous variants in strains with an average genomic coverage > 10. Two structural variants and three SNPs were identified in suppressor genes by visual inspection of the aligned reads (in strains ES036, ES363, ES416, ES943, and ES1163). All whole-genome sequencing data are publicly available at NCBI's Sequence Read Archive (http://www.ncbi.nlm.nih.gov/sra), under accession number PRJNA521449. Detected SNPs and structural variants are listed in Datasets EV5 and EV6.

#### Aneuploidy and ploidy assessment

Qualimap 2 (Okonechnikov et al, 2016) was used to detect (partial) aneuploidies based on variation in sequencing read depth across chromosomes or genomic regions (Fig 5A and B; Appendix Figs S3 and S4A and B; and Dataset EV7). Because the relative increase in coverage caused by an aneuploidy depends on the overall ploidy

(a disomy in a haploid strain will have on average twice as many mapped reads as an euploid chromosome, while a trisomy in a diploid strain will have on average 1.5× as many mapped reads as an euploid chromosome), we analyzed all suppressor strains by flow cytometry to determine ploidy. Briefly, cells were collected from 50 μl of saturated culture and fixed in 70% EtOH for 15 min at room temperature. The fixed cells were washed with water and subsequently treated with RNase A (400 μg/ml, 2 h, 37°C) and proteinase K (2 mg/ml, 1 h, 50°C). Treated cells were washed with 200 mM Tris–HCl (pH 7.5) and stained with 2× SYBR Green (Life Technologies) in 50 mM Tris–HCl (pH 7.5). Stained cells were sonicated and analyzed by flow cytometry using a Becton Dickinson FACSCalibur. Data were analyzed using FlowJo Flow Cytometry Analysis Software, and DNA content was compared to known haploid (BY4741) and diploid (BY4743) controls. Normalized average read depth per chromosome or genomic region values were corrected based on the observed DNA content so that the average normalized read depth of a genomic region in a diploid strain was twice that of a haploid strain.

Total genome size (Dataset EV2) was calculated as the sum of the sizes of all nuclear chromosomes, thereby taking full and partial aneuploidies into account, but disregarding amplifications that occurred in telomeric regions or that affected the *RDN1* locus.

### Aneuploidy complementation screens

For suppressor strains that were found to carry an extra copy of a chromosome (i.e., a disomic chromosome in a haploid strain or a trisomic chromosome in a diploid strain), all genes on the aneuploid chromosome were individually tested for suppression of a deletion allele of the corresponding essential query gene (Appendix Fig S4C). In ~ 6,000 individual transformations, we introduced 2μ plasmids from the MoBY-ORF 2.0 collection (Magtanong *et al*, 2011), each expressing a wild-type copy of a different gene under control of its native promoter, into yeast strain Y7092 (Dataset EV14), thereby creating a collection of array strains each overexpressing another defined gene. We used the MoBY-ORF 2.0 collection because this was the only available systematic library with an appropriate selection marker. The resulting strain collection was crossed into the TS-allele-on-plasmid parental strain of the suppressor strain, thus without the aneuploidy, but deleted for the essential query gene and carrying a TS allele of the query gene on plasmid. When necessary, the mating type of the parental strain was switched before mating. Diploids were subsequently selected, driven through meiosis, and *MAT*a haploid progeny carrying the essential gene deletion allele, the plasmid carrying the TS allele, and the MoBY-ORF 2.0 overexpression plasmid were isolated using the relevant selection markers and the SGA markers that were present in Y7092 (Dataset EV14). The resulting haploid progeny were pinned onto selective media containing 5-FOA, grown for 2 days at 30°C, and pinned onto 5-FOA media again for stronger selection of cells lacking *URA3*, and the colonies were imaged after 4 days at 30°C. Colony size was measured as pixel area (Wagih & Parts, 2014). We determined a *Z*-score and associated *P* value for each query–array gene pair, by calculating how many standard deviations the median size of the query colonies overexpressing the array gene differed from the median size of query colonies carrying an empty vector. We called an array gene a hit when the *Z*-score was > 1.5 and the associated *P* value < 0.05 (Dataset EV10). Genes that were a hit in three or more

screens were classified as frequent flyers and removed from the hit list. Also, plasmids that carried the query gene itself were excluded as hits. The resulting initial screening hits were validated by individual transformations (Dataset EV10). Ten out of 204 tested hits confirmed (5%, Datasets EV2 and EV10).

### Genetic validation of candidate suppressor genes

Candidate suppressor genes were validated as described previously (Van Leeuwen *et al*, 2016). Briefly, each suppressor strain was subjected to three genetic crosses, followed by tetrad analysis of the meiotic progeny of the resulting diploid (Appendix Fig S1B): (i) a cross to a wild-type strain to test for proper 2:2 segregation of the suppressor mutation, i.e., half of the spores carrying a deletion allele of the essential query gene are expected to be dead, while the other half are expected to be suppressed and survive; (ii) a cross to a strain deleted for a gene genetically linked to a suppressor ("neighbor") to test for proper linkage, i.e., all spores carrying both the query mutant and the neighbor deletion allele are expected to be dead, and all spores carrying the query mutation but not the neighbor deletion are expected to be suppressed; and (iii) a cross to a strain carrying a deletion or conditional allele of the suppressor gene. In this last case, if the suppressor mutation was a loss-of-function mutation, all spores carrying the query mutation are expected to be suppressed.

Additionally, the suppressor strains were transformed with plasmids either carrying the wild-type allele of the suppressor gene or an empty vector control (Appendix Fig S1B). Either high-copy plasmids driving genes from their own promoter (Magtanong *et al*, 2011) or low-copy plasmids using the *GAL1*-promoter (Hu *et al*, 2007) were used. If the suppressor mutation is recessive or semi-dominant, overexpression of the wild-type allele of the suppressor gene is expected to reverse the suppression and reduce the fitness of the suppressor strain. Each plasmid was transformed into a wild-type strain as well, to make sure overexpression of the gene does not cause dosage lethality.

Lastly, we directly introduced 17 potential suppressor alleles into a diploid strain that was heterozygous for the corresponding query deletion allele (Appendix Fig S1B). We either amplified the genes carrying the suppressor mutation and a selection marker flanked by appropriate homology regions by PCR, and cotransformed the PCR fragments into the corresponding query gene mutant strain from the BY4743 heterozygous deletion mutant collection (*MAT*a/α *xxxΔ*:: *kanMX4/XXX his3Δ1/his3Δ1 leu2Δ0/leu2Δ0 ura3Δ0/ura3Δ0 met15Δ0/MET15 lys2Δ0/LYS2*; Euroscarf), or we deleted one copy of the suppressor gene in the heterozygous query mutant strain. The diploids were sporulated and subjected to random sporulation analysis to determine whether the introduced mutations could suppress the lethality associated with the query gene deletion allele. Dataset EV2 contains a summary of the results of each of these assays, as well as details on the assignment of mutations as either loss-of-function or gain-of-function variants.

### Suppressor interaction conservation

To test the conservation of bypass suppressor interactions across different genetic backgrounds (Fig 6B), we selected 13 bypass suppressor interactions in which the suppressor mutation involved a complete loss-of-function event (Dataset EV11), and three wild yeast strains with various levels of genetic divergence from the

laboratory yeast strain S288c (FIMA_3, 0.35% divergence; ZP_611, 0.48% divergence; SX3, 0.91% divergence) (Peter *et al*, 2018). First, the query genes were deleted individually in the homozygous diploid wild strains or a S288c control (BY4743) by targeting the query gene with a gRNA in the presence of Cas9 and a kanMX template flanked by appropriate homology regions for replacing the query gene by recombination. Next, the suppressor gene was deleted using a similar strategy with a natMX template. The resulting strains were sporulated and dissected.

If the query gene is essential in the wild yeast strain, homozygous deletion of the gene in a diploid strain should be lethal. If the query gene was homozygously deleted in a diploid wild strain, or if viable haploid progeny were obtained that were deleted for the query gene but not for the suppressor gene, we concluded that the query gene was nonessential in the used genetic background. If the query gene deletion was heterozygous and all viable haploid progeny deleted for the query gene were also deleted for the suppressor gene, we concluded the suppressor interaction was conserved. If no viable haploid progeny lacking the query gene were obtained, we concluded the query gene was essential and not suppressed by deletion of the suppressor gene identified in S288c.

## Computational analysis

### Essential gene list
To define the set of yeast essential genes (Dataset EV13), phenotype data were downloaded from SGD (http://www.yeastgenome.org) on July 7, 2017. "Viable" (nonessential) or "inviable" (essential) annotations were extracted for null (deletion) alleles in haploid S288c strains. For all uncharacterized and verified ORFs that did not have such an annotation, we searched the phenotype data to see whether a deletion allele had been used for these genes in haploid S288c strains. If so, the gene was labeled as nonessential. We manually went through all cases in which annotations were contradictory. Genes that when deleted required supplements for viability were labeled as essential, genes that when deleted only led to lethality under specific conditions as nonessential, and all others as "contradictory". Finally, 133 genes for which a deletion strain was available in the SGA nonessential gene deletion mutant collection (Costanzo *et al*, 2010), but that had no viability data in SGD phenotype dataset, were labeled as nonessential genes.

### Bypass suppression interactions described in the literature
To define a set of bypass interactions that were previously identified in *S. cerevisiae* (Dataset EV3), we made use of a list of manually curated suppression interactions described previously (Van Leeuwen *et al*, 2016) and selected a subset of interactions that met the following criteria: (i) The query gene was essential; (ii) the query gene was either deleted or disrupted; and (iii) the interaction was not identified under specific conditions. We combined this list with the set of "evolvable" essential genes identified by Liu *et al* (2015) and the dispensable essential genes identified by Chen *et al* (2016a).

### Saturation analysis
We performed two types of saturation analysis (Figs 1B and 4B). First, we evaluated if by performing more independent suppressor isolation experiments, we could have identified additional

dispensable essential genes. To be able to do this, we split our experimental data into four artificial screens. For each query strain, we randomly assigned each performed suppressor isolation experiment to one of the four screens. We disregarded query strains with three or fewer experiments, and for query strains with five or more experiments, we randomly selected four experiments. For the first artificial screen, we counted the number of dispensable essential query genes that were identified. For the subsequent screens, we counted the number of identified dispensable essential query genes that were not identified by the previous screens. We repeated this process 1,000 times and calculated the average number of new dispensable essential genes found in each additional screen. Next, we fit a logarithmic model to these average values and used this model to estimate the expected number of novel dispensable essential genes that we would find in additional screens.

In addition, we evaluated whether more independent suppressor isolation experiments were likely to identify additional suppressor genes. As explained above, we randomly assigned each performed suppressor isolation experiment to four artificial screens. For the first artificial screen, we counted the number of dispensable essential query genes with an identified suppressor (i.e., query–suppressor pairs). Importantly, this did not include dispensable essential query genes for which the identity of the suppressor gene could not be established. For the subsequent screens, we counted the number of query–suppressor pairs that were not identified by the previous screens. We repeated this process 1,000 times and calculated the average number of new query–suppressor pairs found in each additional screen. Next, we fit a logarithmic model to these average values and used it to estimate the number of novel query–suppressor pairs that we would find in additional screens.

### Analysis of functional relatedness and enrichment
Functional relatedness between suppression interaction pairs (Figs 3C, F and G, 4A and D) was largely assessed as described previously (Van Leeuwen *et al*, 2016). Briefly, query–suppressor gene pairs were considered functionally related if they shared a biological process GO term annotation (Myers *et al*, 2006; Costanzo *et al*, 2016), had a MEFIT coexpression score > 1 (Huttenhower *et al*, 2006), shared a subcellular localization (Huh *et al*, 2003), or shared a KEGG pathway annotation (Kanehisa *et al*, 2016). Importantly, the set of GO biological process terms was manually curated to disregard broad terms that could result in less functionally relevant coannotation associations (Costanzo *et al*, 2016). While we previously used protein complex annotation data from multiple sources, for all protein complex analyses in the current paper, we used data from the Complex Portal (Meldal *et al*, 2015) (downloaded June 6, 2018). Like before (Van Leeuwen *et al*, 2016), gene pairs that were part of the same protein complex were considered as cocomplexed, and gene pairs in distinct nonoverlapping protein complexes were considered as not cocomplexed. In all cases, only gene pairs for which functional data were available for both the query and the suppressor gene were considered.

For each of these measures of functional relatedness, the expected overlap by chance was calculated by considering all possible pairs between a background set of queries and suppressors. The background set of query genes consisted of the set of dispensable essential query genes (Dataset EV1). As background set for the suppressor genes, we considered all genes in the genome. Pairs in

the suppression interaction dataset were removed from the background set. For a given functional standard, we defined as fold enrichment the ratio between the overlap with the suppression interaction data and the overlap of the background set of pairs with that standard. Significance of the overlap was assessed by Fisher's exact tests.

For the analysis of enrichment of gene sets for different biological processes (Fig 1C), genes were assigned to broadly defined functional gene sets (Van Leeuwen *et al*, 2016). Highly pleiotropic or poorly characterized genes were excluded from the analysis, as were functional categories to which only very few genes were assigned (e.g., "peroxisome" or "drug transport"). Significant enrichment was determined by Fisher's exact test, comparing the observed to the expected proportion of genes in each functional category.

For the analysis of enrichment of dispensable essential genes for other gene- or protein-level properties (Fig 1D), we compared the feature values of dispensable genes to those of indispensable essential genes. These features included 3 binary (having a paralog, coding for a membrane-associated protein, and coding for a protein complex member) and 9 continuous values (dN/dS, sequence length, expression level, expression variation, coexpression degree (the number of genes that share similar expression patterns with a gene of interest), protein disorder, multifunctionality, cocomplex degree (the number of proteins that share a complex with a protein of interest), and the number of complexes a protein belongs to). The sources for these datasets were as follows: genes with paralogs (YeastMine, downloaded Jan. 11, 2018) (Balakrishnan *et al*, 2012), list of membrane-associated proteins (Babu *et al*, 2012), protein complexes (the Complex Portal, downloaded June 6, 2018) (Meldal *et al*, 2015), dN/dS (Koch *et al*, 2012), expression level (Lipson *et al*, 2009), expression variation (Gasch *et al*, 2000), coexpression degree (number of gene partners with a coexpression score > 1) (Huttenhower *et al*, 2006), protein disorder (Oates *et al*, 2013), and multifunctionality (Koch *et al*, 2012). To compute the statistics, we performed Fisher's exact tests for the binary features and Mann–Whitney *U*-tests for the continuous features. Additionally, we evaluated the power of each of these features to predict known dispensable genes by computing their area under the ROC curve (AUROC) and by calculating the deviation from the expected AUROC by chance (0.5).

Data from the Complex Portal (downloaded June 6, 2018) (Meldal *et al*, 2015) were also used for the complex dispensability analysis (Fig 1E, Dataset EV4), and the list of genes with paralogs (YeastMine, downloaded Jan. 11, 2018) (Balakrishnan *et al*, 2012) was also used for the analysis in Appendix Fig S2B.

### Functional impact of suppressor and passenger mutations

For suppressor strains that were sequenced at a coverage of 10× or more, we defined passenger mutations as all SNPs and indels that were present in the strain, but not located in the query or in the suppressor gene (Dataset EV5). The potential functional impact of suppressor and passenger mutations (Appendix Fig S2A) was assessed as described previously (Van Leeuwen *et al*, 2016). Briefly, (i) the deleteriousness of mutations was computed by SIFT (Vaser *et al*, 2016), in which scores below 0.05 are predicted to be deleterious. (ii) The fraction of mutations at protein–protein interaction interfaces was computed using version 2019_01 of Interactome3D for *S. cerevisiae* (Mosca *et al*, 2013). (iii) The fraction of mutations

that occur in disordered regions was calculated using disorder predictions by VSL2b (Peng *et al*, 2006). (iv) The fraction of mutations that occur in essential genes was calculated. For each of these analyses, only missense mutations were considered.

### Multifunctionality

Query genes that were annotated as "highly pleiotropic" or that were annotated to two or more biological processes using broadly defined functional gene sets (Van Leeuwen *et al*, 2016) were considered multifunctional.

### Suppressor gene prediction

For suppressor strains that were found to carry an extra copy of a chromosome (i.e., a disomic chromosome in a haploid strain or a trisomic chromosome in a diploid strain), we predicted the potential causal suppressor genes by ranking the genes in the aneuploidies by their functional relationship to the query gene (Fig 5B and D, Dataset EV9). Specifically, we evaluated the following functional relationships in this order of priority: cocomplex (highest priority), copathway, coexpression, and colocalization (lowest priority). Thus, genes with cocomplex relationships were ranked above those with only copathway relationships. Additionally, the order between genes within a given set was established by evaluating the rest of the functional relationships. For instance, the set of genes that were coexpressed with the query gene, but not in the same complex or pathway, was further ranked by whether they colocalized (highest rank) or not (lowest rank) with the query. See the section "Analysis of functional relatedness and enrichment" for details on the datasets.

### Evolutionary analysis

We evaluated whether dispensable essential genes exhibited different evolutionary properties compared to indispensable essential genes by taking into account the conservation, duplication, and essentiality of gene orthologs in the species *Candida albicans*, *S. pombe*, *Caenorhabditis elegans*, and *Homo sapiens* cell lines KBM7/HAP1 (Fig 6C and D, Appendix Fig S5). We used PANTHER version 15 (Mi *et al*, 2019) to map orthology relationships of dispensable and indispensable essential genes across the analyzed species. For a given gene and species, we considered it to be absent if PANTHER could not find an ortholog in that species, and duplicated if PANTHER found more than one ortholog in that species (including 1-to-many and many-to-many orthology relationships). For conserved genes with a 1-to-1 orthology relationship (i.e., conserved and not duplicated), we evaluated their essentiality in the target species as follows. For *S. pombe*, we defined as essential genes those with an associated lethal phenotype (data downloaded from PomBase in July 2016) (Lock *et al*, 2018) and as nonessential genes those with a viable phenotype. For *C. albicans*, we followed the classification of a recent study (Segal *et al*, 2018). For *C. elegans*, we defined as essential genes those with mutants or RNAi experiments associated with a lethal phenotype at any developmental stage (data downloaded from WormBase in December 2018) (Harris *et al*, 2020) and as nonessential the rest of the genes. For human, we defined as essential genes those found to be essential in the two related human cell lines KBM7 and HAP1 (Blomen *et al*, 2015) and as nonessential the rest of tested genes. Next, for each tested query gene, we counted the number of species in which it was conserved

with a 1-to-1 essential ortholog. Other orthology mapping tools (InParanoid, version 8 (Sonnhammer & Ostlund, 2015); Metaphors, release 2016.01 (Chorostecki *et al*, 2020); and PhylomeDB, yeast phylome ID 515 (Huerta-Cepas *et al*, 2014)) gave similar results (Appendix Fig S5B).

For the comparison of essential gene sets between *S. cerevisiae* strains S288c and Sigma1278b (Appendix Fig S5A), we used data from Dowell *et al* (2010).

To determine human gene essentiality across multiple cell lines (Fig 6E, Appendix Fig S6A and B), we downloaded Achilles dataset 20Q1 from the DepMap portal on February 26, 2020, which contained the results of CRISPR-Cas9 knockout screens for 18,333 genes in 739 cancer cell lines (Meyers *et al*, 2017). For each cell line, we considered as essential genes those with a CERES score below −0.7. We defined genes as indispensable essential in human if they were essential in > 90% of the cancer cell lines, and as nonessential in human if they were essential in < 10% of the cell lines. We defined the rest of genes as context-dependent essential. To show that our results were not dependent on specific cutoffs, we also used a threshold of −0.5 to define essentiality in each cell line and defined genes as nonessential in human if they were essential in < 50% of the cell lines. We evaluated the conservation of essentiality between yeast and human genes by mapping 1-to-1 orthologs using PANTHER version 15 (Mi *et al*, 2019).

To compare the properties of dispensable essential and indispensable essential genes in human, we used a panel of gene features (Fig 6F, Appendix Fig S6C and D). Coexpressed gene pairs were downloaded from SEEK (Zhu *et al*, 2015). We defined as membrane-associated proteins those annotated to the GO term "integral component of membrane" (GO:0016021) and as paralogs those listed in the Duplicated Genes Database (Ouedraogo *et al*, 2012). We used protein complex data defined by CORUM (Giurgiu *et al*, 2019). For the numeric features (coexpression degree, the number of complexes a protein belongs to, and cocomplex degree), we used the values of the indispensable essential genes to perform a *Z*-score normalization of the values of the context-dependent essential genes. Instead of using the mean value of the indispensable essential genes, we used the median. Statistical significance was evaluated by Mann–Whitney *U*-tests. For binary features (membrane-associated proteins, genes with paralogs, and complex membership), we compared the fraction of context-dependent essential genes to the fraction of indispensable essential genes that displayed that particular feature. Statistical significance was evaluated by Fisher's exact tests.

For each human gene, we estimated their presence or absence in 11 other species by using PANTHER version 15 orthology relationships (Mi *et al*, 2019) (Appendix Fig S6E). For each species, we compared the fraction of context-dependent essential genes in human to the fraction of indispensable essential genes in human that did not have an ortholog in that species. Statistical significance was evaluated by Fisher's exact tests.

### Dispensable essential gene prediction

To predict gene dispensability for essential genes (Fig 7, Appendix Fig S7, Dataset EV12), we used a panel of gene features, evolutionary features, and gene function information. The evolutionary features included the absence and duplication of genes, and essentiality data of 1-to-1 orthologs in *C. albicans*, *S. pombe*,

*C. elegans*, and human cell lines (see the section "Evolutionary analysis"). We calculated orthology relationships using PANTHER (Mi *et al*, 2019) as explained above, except that we used version 9 instead of 15. Note that the different version of PANTHER does not substantially affect our predictor (Spearman's correlation = 0.95 between prediction scores). As gene features, we used coexpression degree (number of gene partners with a coexpression score > 1) (Huttenhower *et al*, 2006), protein disorder (Oates *et al*, 2013), dN/dS (Koch *et al*, 2012), expression variance in response to genetic (Brem & Kruglyak, 2005) or environmental (Gasch *et al*, 2000) perturbations, protein length, multifunctionality (see the section "Multifunctionality"), PPI degree (Koch *et al*, 2012), transcript count (Lipson *et al*, 2009), protein half-life (Christiano *et al*, 2014), protein abundance (Ho *et al*, 2018), the number of pfam domains (Finn *et al*, 2016), the number of complexes in which a gene product participates, cocomplex degree (the number of proteins that share a complex with a protein of interest), and whether a gene has a duplicate (YeastMine, downloaded January 11, 2018) (Balakrishnan *et al*, 2012), encodes for a membrane-associated protein (Babu *et al*, 2012), or a protein in a complex. Complex data were downloaded from the Complex Portal (downloaded June 6, 2018) (Meldal *et al*, 2015). We performed *Z*-score normalization of the numeric features as explained above. Finally, broadly defined functional gene sets (Van Leeuwen *et al*, 2016) were used to specify gene function.

We used the R package "randomForest" (Liaw & Wiener, 2002) to train a random forest classifier with class-balanced subsets by undersampling indispensable genes. Performance of the predictor was evaluated with out-of-bag data that were not used for training (36 and 81% of the dispensable and indispensable genes, respectively) and two datasets available in the literature (Dataset EV3) (Liu *et al*, 2015; Van Leeuwen *et al*, 2016). We repeated the training process after removing one variable at a time and found the result to be very robust with AUCs ranging from 0.72 to 0.75. Out of all the included features, the essentiality of a gene in *S. pombe* was affecting the performance of the random forest the most.

Thirteen essential genes with a high dispensability prediction score and 15 genes with low dispensability prediction scores were experimentally tested for dispensability using the methods described in the "TS-allele-on-plasmid collection construction" and "Bypass suppressor isolation" sections above, with the exception that only two independent suppressor isolation attempts were performed per gene (Fig 7B, Dataset EV12). These genes were selected based on the availability of TS alleles, which are needed to construct the query strains.

### Classifying essential genes as either dispensable or core essential

To define a core set of essential genes (Dataset EV13), we selected essential genes that were either indispensable in our experiments (Datasets EV1 and EV12) or that were not experimentally tested in our assay but that we predicted to be indispensable with a score below 0.5 (see the previous section "Dispensable essential gene prediction"). Based on the number of dispensable genes we experimentally identified with a low predicted dispensability score, we estimate that at this cutoff, the actual probability of the gene being indispensable is ~ 89%. In both cases, we removed essential genes from the list of core essential genes if bypass suppressors had been described in the literature (Dataset EV3).

All 131 essential genes that could be bypassed in our experiments (Datasets EV2 and EV12) were classified as dispensable essential (Dataset EV13). An additional 69 genes that were not experimentally tested in our assay but that we predicted to be dispensable with a score above 0.5 were classified as dispensable essential. At this cutoff, we estimate that 38% of the predicted dispensable genes is actually dispensable. All other genes were not further classified (Dataset EV13).

## Data availability

The data produced in this study are available in the following database:
- Whole-genome sequencing data: NCBI's Sequence Read Archive, accession number PRJNA521449 (https://www.ncbi.nlm.nih.gov/bioproject/PRJNA521449/).

**Expanded View** for this article is available online.

## Acknowledgements
We thank M. Costanzo, A. Batté, B. Ünlü, T. Sing, M. Hung, C. Ross, the Donnelly Sequencing Centre, and the Faculty of Medicine Flow Cytometry Facility for critical reading of the manuscript and technical assistance. We also thank J. Rine and A. Warren for discussion of bypass suppressor mechanisms. This work was supported by grants from the Canadian Institutes of Health Research (FDN-143264 and FDN-143265) (C.B., B.J.A.), the National Institutes of Health (R01HG005853) (C.B., B.J.A., C.L.M.), the Swiss National Science Foundation (PCEGP3_181242) (J.v.L), and a Ramon y Cajal fellowship (RYC-2017-22959) (C.P.). C.B., B.J.A., F.P.R., and C.L.M. are Senior Fellows in the Canadian Institute for Advanced Research, Genetic Networks Program.

## Author contributions
JvL, CB, and BJA supervised the project. CLM, FPR, and PA supervised specific experiments and analyses. JvL, GT, ZW, JH, MG, WL, ES, ZL, ML, and ADSL conducted experiments. CP performed the computational and statistical analyses. JvL, CP, ZW, JH, JW, MU, and NvL analyzed and interpreted the data. JvL, CP, BJA, and CB prepared the manuscript.

## Conflict of interest
The authors declare that they have no conflict of interest.

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
