## [Review Process File · Molecular Systems Biology]

Systematic analysis of bypass suppression of essential genes

Jolanda Van Leeuwen, Carles Pons, Guihong Tan, Jason Wang, Jing Hou, Jochen Weile, Marinella Gebbia, Wendy Liang, Ermira Shuteriqi, Zhijian Li, Maykel Lopes, Matej Usaj, Andreia Dos Santos Lopes, Natascha van Lieshout, Chad Myers, Frederick Roth, Patrick Aloy, Brenda J. Andrews, and Charles Boone

DOI: 10.15252/msb.20209828

Corresponding author(s): Charles Boone (charlie.boone@utoronto.ca) , Brenda J. Andrews (brenda.andrews@utoronto.ca), Charles Boone (charlie.boone@utoronto.ca), Jolanda Van Leeuwen (jolanda.vanleeuwen@unil.ch)

Review Timeline:

Submission Date:	1st Jul 20
Editorial Decision:	3rd Aug 20
Revision Received:	11th Aug 20
Accepted:	13th Aug 20

Editor: Maria Polychronidou

Transaction Report:

Thank you again for submitting your work to Molecular Systems Biology. We have now heard back from the three referees who agreed to evaluate your study. Overall, the reviewers are rather supportive. However, they raise a series of concerns, which we would ask you to address in a revision.

I think that the recommendations of the reviewers are rather clear and there is therefore no need to repeat the points listed below. Most issues raised are relatively minor. Please let me know in case you would like to discuss in further detail any of the points raised.

On a more editorial level, we would ask you to address the following issues.

REFeree REPORTS

Reviewer #1:

In this study by van Leeuwen and colleagues, the authors have studied the properties of suppressor mutations of dispensable essential genes. Taking advantage of a large collection of 728 temperature sensitive alleles of essential genes, the authors discovered 124 essential genes that can be suppressed by the acquisition of mutations over a very short time span. They characterize the properties of so called dispensable essential genes, relative to essential genes where suppression is not easily reached by the mutations that are possible within the experiments performed. As in a previous study performed using a different experimental approach (Liu et al, 2015), the authors found that dispensable essential genes have properties that are intermediary between non essential and non dispensable essential genes.

The authors then identified the specific suppressor interactions of which a total of 141 unique bypass interactions were validated based on a series of extensive follow up experiments. The properties of these interactions were studied computationally with some found to be similar to those found in a previous study of the same authors for non essential gene deletions (Van Leeuwen et al, 2016). Most interactions (68%) are found to be between pairs of genes that are functional related, typically involve a single gene and bypass suppressor mutations in protein complexes were overwhelmingly (80%) found to cause gain of function effects. Although most sequence suppressor strains have gene copy number changes only a small number of these events could be directly linked to a suppressor interactions. Perhaps not unexpectedly, the specific suppressor interactions and even the essential genes are not conserved in different *S. cerevisiae* strains for a small subset tested.

Finally, dispensable essential genes were shown to have specific properties that hold true in human cells and can be used as the basis for building predictors.

The work in the manuscript is very extensive and complete and provides a significant advance towards our understanding of gene essentiality and its plasticity. It goes beyond what was previously achieved in Liu et al, 2015 and is quite extensive in how it makes use of the data acquired. I don't have any major concerns and only a few minor suggestions.

- The paper is easy to follow but reads as a list of interesting observations and sometimes it is not easy to get a summary of what was discovered. In particular there is no easy summary of the mechanisms of suppression. It would be great to have some summary of the contents of Table S2 such that one could have a global sense of the fraction of suppressions in regards to: complexity (single genes vs multiple genes), type of mutation (missense variants, copy number changes), effect (gain or loss), maybe others characteristics the authors may find relevant.

- The authors were capable of assigning variants to a gain or loss of function effect for a very large number of sequenced variants. The reasoning behind this assignment is not always clear from reading the methods. Looking through Table 2 there are some cases where I can think through the experiments done to reach the same conclusions but this is not often the case. It would be useful if the authors could add in Table 2 a text description of how each of the calls for gain/loss of function was reached. Were there cases where the loss/gain annotation was predicted but not experimentally confirmed ?

- The authors found a higher functional relation between suppressor interactions of essential genes than non essential genes. This could be because essential genes are more studied. I don't suspect this is the case but the authors could make sure by using metrics of gene-gene functional relatedness that are not socially biased such as high throughput studies of protein interactions.

- Very minor but the definition of functional related pairs may be too broad such as if all it takes is belonging to the same GO term. This has no impact on the conclusion that pairs of genes in a suppressive interaction are significantly more likely to functionally interact. Maybe the authors could add to the methods some description of the steps taken to avoid using extremely broad GO terms.

- I accept that this is beyond the scope of this current project but there are several proteins with multiple gain/loss of function missense variants annotated to them based on this work. From a protein structural perspective, there could be interesting novel findings, in particular for some of the missense mutations that cause gain of function effects. Maybe something that the authors could look into in the future.

Reviewer #2:

In this paper, van Leeuwen and colleagues examine the extent to which essential genes might in fact be dispensable, given other genetic changes. They find a large number of dispensable essential genes, and tease apart genetic and molecular mechanisms that enable bypass suppression of these genes. The authors show that different types of genetic changes, such as aneuploidies/diploidization, loss-of-function mutations, and gain-of-function mutations, can rewire

the ways that biological systems work, rendering certain essential genes non-essential.

In my opinion, this was a very interesting paper. The science is rigorous, the text is clear, and the figures are aesthetically pleasing. This paper might be viewed as dense by some, but I thought the authors did a nice job distilling a substantial amount of information down into a relatively succinct manuscript. This paper will likely be of interest to a broad range of scientists focused on genetics, systems biology, and evolution.

My comments are mostly minor:

- 'Here, we describe the construction of a collection of haploid yeast strains carrying deletion alleles of most essential genes.' I had to read this sentence a few times because it sounds like the strains might carry multiple deletion alleles. Wording like 'single' or 'individual' might help.

- What is not entirely clear in the main text and Figure 1 is how the haploid deletions are generated in the presence of the ts plasmids. After reading the Methods, I fully understood the approach. However, I wonder if a bit more technical information should be provided in the Results section of the text, as well as Figure 1.

- In the INO80 anecdote, it sounds a bit like INO80 is a HAT, but that is not the case. Perhaps the wording could be a little clearer regarding INO80's function.

- I felt more attention could be paid to the comparison of conditional essentiality and dispensable essentiality. If the genetic changes that render certain essential genes dispensable were already present, then conditional essentiality would be observed instead. It seems like these concepts are closely related, if not different manifestations of the same phenomenon. I am sure the authors have some good insights into this matter.

Reviewer #3:

Summary

The manuscript entitled 'Systematic analysis of bypass suppression of essential genes' by van Leeuwen et al describes a thorough analysis of genetic suppression in yeast to alleviate cell viability defects caused by inactivation of essential genes. This is an extensive experimental study with impressive analysis of the obtained data. A first important finding is that about 17% of the essential genes can be bypassed by suppressor mutations and that these bypassable genes (termed dispensable essential genes) have properties that are more typical for non-essential genes. The systematic analysis of this type of genetic interaction leads to wonderful examples such as the recognition of 11 protein complexes for which all essential subunits could be bypassed whereas in other cases this only applied to a specific submodule. The work continues with characterization of the suppressor mutants. Here, key observations were that most bypassable essential genes could only be suppressed by a single genetic mechanism (for me, this was unexpected and important) and that most of the suppression events involved gene activation. Figure 2 highlights the organization of bypass suppression and is arguably the most important figure of this work. The dispensable essential genes are less often essential in other organisms highlighting the occurrence of suppression during evolution. Ultimately, the authors make efforts to predict dispensable essential genes and to predict suppressors on aneuploid chromosomes.

General remarks

This is a wonderful and extensive study on genetic suppression related to cell viability (which is a clear type of genetic interaction). The systematic aspect of this work therefore provides a good conceptual insight in this type of genetic interaction, the different mechanisms behind it and the relevance for gene essentiality in different organisms. The completeness and the analysis make this work is also a significant advance in relation to the previous work (van Leeuwen et al, Science 2016). This work would be of relevance for those interested in genetic interactions in general and scientists interested in yeast genetics or essential genes. The systematic scope of this work should be of interest for readers of Molecular Systems Biology.

Major points

The authors provide such an amount of data that it almost becomes too much. The high number of examples given and numbers of genes that belong to different categories makes it truly difficult to read. Crucially, this anecdotal style detracts from the very important conclusions that can be drawn from this work (also listed in the summary: 1. a key fig 2 depicting virtually all suppression mechanisms for essential yeast genes, 2. that bypassable genes have different properties, 3. are less often essential in other organisms and 4. can typically be bypassed by only one genetic event, and 5. often suppression occurs through gene activation). In my view the authors should consider to focus more on these important conclusions, while still providing the most interesting examples and highlighting the systematic experimental aspect of their work.

Minor points

P4: the paragraph starting with 'remains unknown (Table S3)' is a bit unclear. 4 genes have bypass suppressor mutants in S288c and were not found: how does this fit with the conclusion that more dispensible essential genes could not be found?

P6: 'suppressor mutations of nonessential deletion mutants': perhaps explain here which traits were suppressed?

Editor's comments:

We have made all the changes suggested by the editor, most importantly by providing keywords, text for the synopsis, and author contributions, and by replacing the supplementary information with the appendix and expanded view format.

Reviewer comments:

We thank all Reviewers for their positive and constructive comments on our manuscript. We appreciate that our paper describes a large amount of data and, as suggested by Reviewers #1 and #3, we have made several revisions to emphasize the main findings and conclusions of our study. Specific comments are addressed below.

Reviewer #1:

1. The paper is easy to follow but reads as a list of interesting observations and sometimes it is not easy to get a summary of what was discovered. In particular there is no easy summary of the mechanisms of suppression. It would great to have some summary of the contents of Table S2 such that one could have a global sense of the fraction of suppressions in regards to: complexity (single genes vs multiple genes), type of mutation (missense variants, copy number changes), effect (gain or loss), maybe others characteristics the authors may find relevant.

We agree with the reviewer that some of the main findings from our study about mechanisms of suppression may have been obscured somewhat by the discussion of specific examples of bypass suppression. To address this issue, we have included pie charts summarizing the various characteristics of the suppressors in Figure 3 (the new Fig 3A and B). To provide more focus on our main results, we have also slightly reorganized the sections describing the suppressors, and have included summary sentences at the end of each section. Specifically, we made the following changes to the text:

- We merged the “Properties of bypass suppressors of essential gene deletion mutants” and “Mechanistic categories of suppression interactions” sections.
- On page 8, we added: “Thus, bypass suppressors of essential gene deletion mutants share several properties with suppressors of nonessential gene deletion mutants, such as a strong functional connection between the query and the suppressor gene. However, essential gene bypass suppressors more frequently involve gain-of-function mutations in essential suppressor genes or in genes encoding members of the same complex as the query gene.”
- On page 10, we added: “To summarize, in cases where multiple suppressor mutations co-occur in a suppressor strain, either both mutations may be required for the bypass suppression phenotype, or one suppressor mutation may act as a bypass suppressor and the second mutation further improves the fitness of the suppressor strain.”

2. The authors were capable of assigning variants to a gain or loss of function effect for a very large number of sequenced variants. The reasoning behind this assignment is not always clear from reading the methods. Looking through Table 2 there are some cases where I can think through the experiments done to reach the same conclusions but this is not often the

case. It would be useful if the authors could add in Table 2 a text description of how each of the calls for gain/loss of function was reached. Were there cases where the loss/gain annotation was predicted but not experimentally confirmed ?

As suggested by the reviewer, we have added a column to Table S2 (now called Dataset EV2), listing for each case the information that was used to categorize mutations as either loss- or gain-of-function mutations. We have not observed any cases where a mutation that was predicted to have a loss-of-function effect (such as a frameshift, or early stop codon mutation) was found to have a gain-of-function phenotype in our experiments, or vice versa.

3. The authors found a higher functional relation between suppressor interactions of essential genes than non essential genes. This could be because essential genes are more studied. I don't suspect this is the case but the authors could make sure by using metrics of gene-gene functional relatedness that are not socially biased such as high throughout studies of protein interactions.

We did not see a significant difference in the fraction of suppressor-query gene pairs that share a biological process annotation between essential and nonessential query gene deletion mutants when we use a complete, unbiased set of biological process annotations (68% and 65% respectively, Fig 3F (previously Fig 3D)). However, we do observe a higher fraction of essential query genes that share pathway or complex annotation with a suppressor gene when compared to nonessential query genes (20% and 10% respectively, Fig 3F). We had a closer look at the pathway and complex datasets, and noticed that essential genes were indeed about twice as likely to have a pathway or complex annotation than nonessential genes, potentially explaining the observed differences in shared pathway or complex relationships between essential and nonessential query gene-suppressor pairs. We thus removed the following sentence from the manuscript: "... suggesting that suppressors of essential gene deletion mutants tend to have an even closer functional connection to the query gene than the suppressors of nonessential genes (Fig 3F).".

4. Very minor but but the definition of functional related pairs may be too broad such if all it takes it belonging to the same GO term. This has no impact on the conclusion that pairs of genes in a suppressive interaction are significantly more likely to functionally interact. Maybe the authors could add to the methods some description of the steps taken to avoid using extremely broad GO terms.

We agree, and we have indeed only used a subset of more specific GO terms when determining GO coannotation. We have added the following text to the Methods section to clarify this: "Importantly, the set of GO biological process terms was manually curated to disregard broad terms that could result in less functionally relevant co-annotation associations (Costanzo et al, 2016).".

5. I accept that this is beyond the scope of this current project but there are several proteins with multiple gain/loss of function missense variants annotated to them based on this work. From a protein structural perspective, there could be interesting novel findings, in particular for some of the missense mutations that cause gain of function effects. Maybe something that the authors could look into in the future.

We thank the Reviewer for this suggestion. We have done an analysis that is related to this suggestion, in which we looked at the distribution of mutations across suppressor genes for

which multiple independent suppressor mutations were identified in our experiments. We saw that loss-of-function mutations are generally spread over the full length of the gene, whereas gain-of-function mutations tended to cluster in specific domains. Because the number of mutations per gene was often limited, this particular analysis was not included in the current manuscript, but this is definitely something that would warrant further investigation in the future.

Reviewer #2:

1. 'Here, we describe the construction of a collection of haploid yeast strains carrying deletion alleles of most essential genes.' I had to read this sentence a few times because it sounds like the strains might carry multiple deletion alleles. Wording like 'single' or 'individual' might help.

We have changed the text to: “Here, we describe the construction of a collection of haploid yeast strains, each carrying a single deletion allele of a different essential gene.”.

2. What is not entirely clear in the main text and Figure 1 is how the haploid deletions are generated in the presence of the ts plasmids. After reading the Methods, I fully understood the approach. However, I wonder if a bit more technical information should be provided in the Results section of the text, as well as Figure 1.

We have now included in Appendix Figure S1A a diagram showing the complete experimental pipeline from strain construction to bypass suppressor isolation, and we have provided details on the strain construction strategy in the figure legend. We have also added the following to the main text: “To construct these strains, we PCR-amplified TS alleles from available TS strains (Costanzo *et al*, 2016), and cotransformed the PCR product and a linearized plasmid carrying a haploid-selection cassette into a diploid yeast strain that was heterozygous for a deletion allele of the corresponding essential gene. The resulting diploid strains carrying an assembled plasmid were sporulated, and haploid progeny carrying the deletion allele of the essential gene and the TS allele on plasmid were selected using the haploid selection cassette present on the plasmid (Appendix Fig S1A, Materials and Methods).”.

3. In the INO80 anecdote, it sounds a bit like INO80 is a HAT, but that is not the case. Perhaps the wording could be a little clearer regarding INO80's function.

We have clarified this by changing the text to the following: “For example, all three bypass suppressor strains of *INO80*, which encodes a member of the INO80 chromatin remodeling complex involved in the regulation of chromosome segregation (Chambers *et al*, 2012), were diploidized. In this case, suppression occurred via homozygous loss-of-function mutations in histone deacetylase genes (Dataset EV2), which likely counteract the reduced histone acetylation due to histone reorganization in *ino80* mutants (Chambers *et al*, 2012; Papamichos-Chronakis *et al*, 2011).”.

4. I felt more attention could be paid to the comparison of conditional essentiality and dispensable essentiality. If the genetic changes that render certain essential genes dispensable were already present, then conditional essentiality would be observed instead. It

seems like these concepts are closely related, if not different manifestations of the same phenomenon. I am sure the authors have some good insights into this matter.

We agree with the Reviewer that the concepts of dispensable essentiality and conditional essentiality are likely closely related, and we do indeed observe that dispensable essential genes are more frequently nonessential in other yeast strains (Fig 6B, Appendix Fig S5A). However, it remains unknown whether mutations in the same bypass suppressor genes that are driving essential gene dispensability are underlying the conditional essentiality in other genetic backgrounds. We have added the following text to the discussion (page 17): “We showed that dispensable essential yeast genes are often nonessential in other *S. cerevisiae* backgrounds (Fig 6B, Appendix Fig S5A), suggesting that dispensable essentiality and conditional essentiality (i.e. differences in gene essentiality between genetic backgrounds) are closely related, and that bypass suppressors isolated in the lab might reflect suppression events that occur during evolution. However, as most of our bypass suppressor strains have a fitness defect compared to wild-type strains (Dataset EV2), we suspect that multiple suppression variants may be present in the non-reference genetic backgrounds to achieve wild-type fitness in the absence of the conditional essential gene. Indeed, we have previously shown that complex networks of genetic modifiers often underly differences in gene essentiality between two yeast strains (Hou *et al*, 2019), and here we found that multiple suppressors can combine to increase the fitness of the suppressor strain.”

Reviewer #3:

1. The authors provide such an amount of data that it almost becomes too much. The high number of examples given and numbers of genes that belong to different categories makes it truly difficult to read. Crucially, this anecdotal style detracts from the very important conclusions that can be drawn from this work (also listed in the summary: 1. a key fig 2 depicting virtually all suppression mechanisms for essential yeast genes, 2. that bypassable genes have different properties, 3. are less often essential in other organisms and 4. can typically be bypassed by only one genetic event, and 5. often suppression occurs through gene activation). In my view the authors should consider to focus more on these important conclusions, while still providing the most interesting examples and highlighting the systematic experimental aspect of their work.

We agree that sections about the bypass suppressor genes described a high number of categories and numbers of genes (see also Reviewer #1, comment #1). To provide more focus on our main results, we have reorganized these sections, we have added summary sentences at the end of each section, and we have included summary pie charts in Fig 3. Please see Reviewer #1, comment #1 for details.

2. P4: the paragraph starting with 'remains unknown (Table S3)' is a bit unclear. 4 genes have bypass suppressor mutants in S288c and were not found: how does this fit with the conclusion that more dispensible essential genes could not be found?

There are indeed 4 genes (*SEC13*, *NUPI1*, *STT4*, *KARI1*) for which bypass suppressors have been described in the literature, but for which we were unable to isolate any bypass suppressors in our experiments. For these 4 genes, we performed 3-5 independent attempts to isolate suppressors, but failed every time. In all of these cases, the suppressor mutations described in the literature were not particularly rare mutations, and we thus should have been

able to identify them with the number of cells we used in our assay. We thus think that even with repeating the experiment additional times, we will not be able to isolate suppressors for these genes in our experimental assay. We suspect that the differences between our results and those described in literature may be due to variation in environmental conditions (temperature, media), small changes in genetic background between S288c strains in different labs, or potentially unidentified problems with some of the strains. We added the sentence: “These 4 genes may have been missed in our assay due to differences in environmental conditions or slight changes in genetic background between S288c strains from different labs.” to clarify this.

3. P6: 'suppressor mutations of nonessential deletion mutants': perhaps explain here which traits were suppressed?

These were nonessential deletion mutants that had a fitness defect. We clarified the text by adding: “nonessential deletion mutants that displayed a growth defect”.

Thank you again for sending us your revised manuscript. We think the the performed revisions have satisfactorily addressed the issues raised by the reviewers. We are now satisfied with the modifications made and I am pleased to inform you that your paper has been accepted for publication.

Corresponding Author Name: Charles Boone

Manuscript Number: MSB-20-9828R